# DeepFocus: fast focus and astigmatism correction for electron microscopy

P. J. Schubert [1], R. Saxena[1] & J. Kornfeld [1] ✉

High-throughput 2D and 3D scanning electron microscopy, which relies on automation and dependable control algorithms, requires high image quality with minimal human intervention. Classical focus and astigmatism correction algorithms attempt to explicitly model image formation and subsequently aberration correction. Such models often require parameter adjustments by experts when deployed to new microscopes, challenging samples, or imaging conditions to prevent unstable convergence, making them hard to use in practice or unreliable. Here, we introduce DeepFocus, a purely data-driven method for aberration correction in scanning electron microscopy. DeepFocus works under very low signal-to-noise ratio conditions, reduces processing times by more than an order of magnitude compared to the state-of-the-art method, rapidly converges within a large aberration range, and is easily recalibrated to different microscopes or challenging samples.

The high resolution of electron microscopy (EM), and the ability to image every sample detail, for tissue with the help of dense heavy-metal staining, remain unrivaled[1,2]. Massive improvements in automation allows the acquisition of 3D images of biological samples with nanometer resolution spanning millimeters[3,4]. While EM connectomics, the complete mapping of neuronal tissue, has been one of the key applications, automated 3D EM also enabled studies ranging from the analysis of cellular SARS-CoV-2 replication[5] to fuel cell research[6], demonstrating its wide applicability.

A key component of automated (3D) EM is to maintain high-quality images over the entire acquisition process, often involving millions of individual 2D images and 24/7 operations. This renders manual microscope parameter adjustments virtually impossible. Automatic defocus and astigmatism correction algorithms remain a challenge despite their necessity, especially in high-throughput electron microscopy. This can be explained by sample diversity, the tight constraints on algorithm execution time, aberration correction convergence speeds, and low-electron dose budgets to avoid artefacts.

Existing solutions[7–10] in the area of scanning electron microscopy (SEM) are usually based on explicit physical models of the electron beam and its interaction with the sample (Fig. 1a). Measurements (images) with known perturbations are taken, followed by focus and stigmation parameter inference to estimate the wavefront aberrations. These physically grounded approaches, and those employing classical image

sharpness scores[11–13], often struggle with generalization. In other words, they frequently fail to perform well on novel samples without expert parameter tuning. This tuning may be infeasible for users, particularly when the algorithm is integrated into the microscope control software.

A recent study introduced a complex approach that employs two artificial neural networks to evaluate the quality of SEM images and subsequently estimate working distance corrections using an updated state vector and a database comprising tens of thousands of manually labeled images[14]. Reinforcement learning was applied to the problem of electron beam alignment[15] and deep learning models were successfully used for focus correction in light microscopy[16,17]. Motivated by these developments and the strong performance of convolutional neural networks in general image processing tasks, we devised a deep learning-based focusing and stigmatization correction method for scanning electron microscopy. Our algorithm features near-instant inference time, rapid convergence, functionality with low-electron dose noisy images, and a user-friendly process for recalibrating it to new machines and samples without the need for expert knowledge, ensuring convergence in all application scenarios.

## Results

### The DeepFocus model

The image of a flat specimen in a scanning electron microscope is optimally captured when the size of the spot of the electron beam is

[1]Max Planck Institute for Biological Intelligence, Martinsried 82152, Germany. ✉ e-mail: joergen.kornfeld@bi.mpg.de

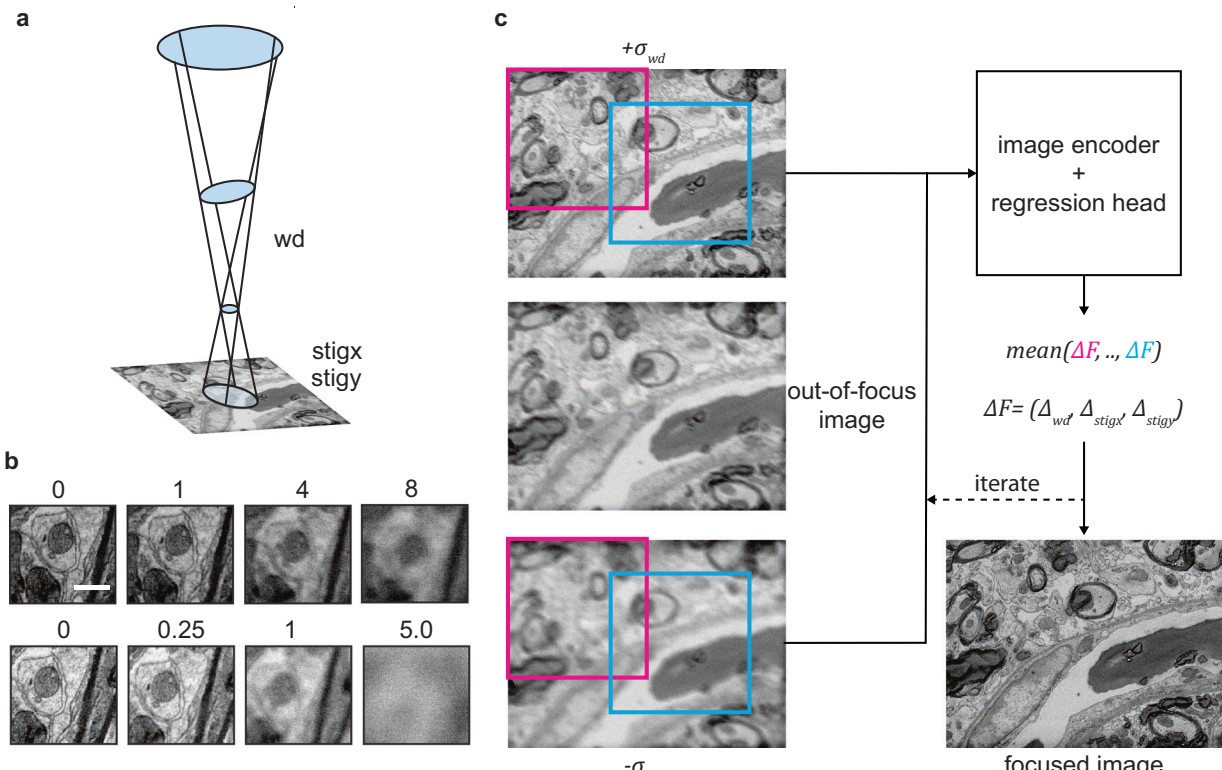

**Fig. 1 | SEM beam formation and DeepFocus algorithm. a** Schematic of the electron beam and the parameters that are controlled by DeepFocus. **b** Defocus - and astigmatism series ($n = 1$) that shows the influence of mild working distance (top row: 0 to 8 μm) and stigmator changes (bottom row: 0 to 5 a.u.) on image quality for a Zeiss Merlin SEM with 800 ns dwell time and 10 nm pixel size. Scale bar is 500 nm. **c** The out-of-focus image (example resolution 1024 × 768, 10 nm pixel size) is perturbed (symmetric perturbation $\sigma_{wd} = \pm 5$ μm) and randomly located patch pairs (shared offset within image pair, e.g. blue and pink squares with shape 512 × 512) of fixed shape are cropped and processed by DeepFocus - an artificial neural network consisting of an image encoder and a regression head. A definable number of independent predictions is used to calculate a correction term $\Delta F$ for each focus parameter (wd: working distance, stig x: stigmator x; stig y: stigmator y). All SEM images have 10 nm pixel size. Scale bar in b is 500 nm.

smaller than the sampling distance. Commonly, three parameters, working distance, on-axis stigmator and diagonal stigmator, henceforth referred to as stig x and y, can be adjusted by SEM operators to directly control the spot shape and bring it below the pixel size at the beam-specimen interaction point (Fig. 1a), consequently leading to sharp image formation (Fig. 1b).

The DeepFocus algorithm takes as input two SEM images with a known working distance perturbation $\sigma_{wd}$ around the current microscope working distance and stigmator settings $F = \left[ f_{wd} f_{stig\,x} f_{stig\,y} \right]$ to estimate the direction and magnitude correction of the beam parameters, exploiting phase diversity[18]. Note, that a single image is not sufficient to estimate the aberrations. Multiple subregions (patches) are cropped from the two perturbed input images (Fig. 1c), and processed independently by a convolutional neural network. This network is trained to infer the $\Delta F$ that leads to a sharp image when added to $F$. The resulting multiple $\Delta F$ estimates, one for each input patch pair, are reduced by a mean operation, which serves as final output for a single iteration.

To assess the model, we trained the network for about 44 h on a single GPU on a set of 32 sample locations with different aberration parameters (in total $n = 320$ input image pairs; Supplementary Fig. 1). We subsequently tested the model on location-aberration pairs that were not part of the training set (Methods). DeepFocus rapidly converges toward the target $\widetilde{\Delta F}$ values within three iterations (Fig. 2a, b; perturbed image electron dose: ~19 electron/nm²), even for low signal-to-noise (SNR) ratio image pairs (Fig. 2c, d; ~5 electron/nm²) and small input patches (Supplementary Fig. 2a; Supplementary Table 1). We also examined the impact of input alignment, a strict requirement for

example for the algorithm by Binding and Denk[7], and found that the model performs well also in the extreme case that the patches in an input pair did not share the same offset, but were chosen randomly (Supplementary Fig. 2b).

The average estimated correction $\Delta F = \left[ \Delta f_{wd} \Delta f_{stig\,x} \Delta f_{stig\,y} \right]$ after a single iteration was assessed at nine distinct locations (evenly spaced grid with an edge length of 100 μm) for an expanded range of initial defocus (working distance perturbation in μm of ± 20, ± 10, ± 5, ± 2, ± 1) to evaluate the model's learned transformation's goodness of fit. The relationship between the target correction for the working distance $\widetilde{\Delta f}_{wd}$ (the negative introduced defocus) and model output $\Delta f_{wd}$ should ideally be linear, specifically, it should follow

$$\Delta f_{wd} = c_1 \times \widetilde{\Delta f}_{wd} + c_2 \tag{1}$$

with $c_1 = 1$ and $c_2 = 0$. Using ordinary least squares (OLS; from the statsmodels Python package) to fit a line resulted in $c1 = 0.9093 \pm 0.006$ and $c2 = 0.3436 \pm 0.061$ (± 1σ interval), signifying a slight yet significant deviation from the identity function. Nevertheless, the model effectively learned to deduce the correction direction. Notably, the stigmator values were barely changed by the model, when just the working distance was perturbed (Fig. 2e, Supplementary Fig. 3). The remaining mean absolute difference of the working distance

$$|\delta_{wd}| = |\Delta f_{wd} - \widetilde{\Delta f}_{wd}| \tag{2}$$

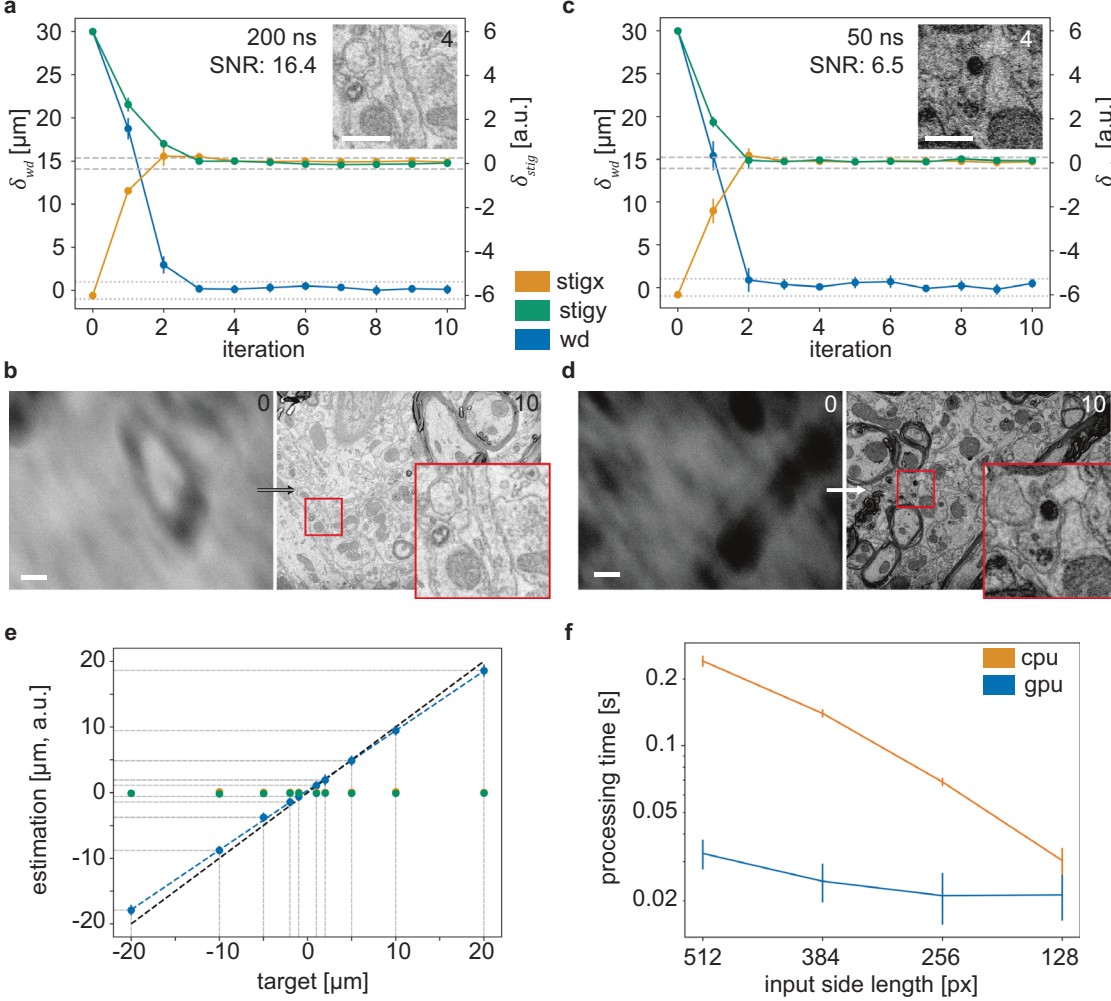

**Fig. 2 | DeepFocus convergence and processing time. a** Convergence plot based on the mean of $n = 5$ predictions on $512 \times 512$ input patch pairs, cut out from two perturbed images acquired with $1024 \times 768$ with 200 ns dwell time. The y-axis shows the difference to the correct focus values (dashed and dotted horizontal lines indicate 0.25 and 1 μm margin of stig and wd, cf. Fig. 1b) with an initial aberration of 30 μm, +6, −6 (wd, stig x, stig y). Numbers in images indicate the iteration. **b** Image ($n = 1$) before and after applying DeepFocus, 800 ns dwell time. **c, d** Same as in a,b but with 50 ns dwell time, including the inset in c. **e** Correction estimate ($n = 9$ different locations; in μm for wd and a.u. for stig x and stig y) after one iteration using 5 input patches ($512 \times 512$) with 200 ns pixel dwell time. Colors correspond to those in a and c. **f** Processing time per input patch pair for different patch side lengths ($n = 10$ repetitions, each with 5 input patch pairs) on the microscope PC. Scale bars are 500 nm in a,c and 1 μm in b,d. Data in all plots are presented as mean values ± standard deviation (SD). Source data are provided as a Source Data file.

was closer to the target value $\Delta\widetilde{f}_{wd}$ for smaller initial deviations (Supplementary Fig. 3). This, combined with the correct direction, enables convergence.

Apart from the ability of correcting image aberrations with high accuracy, a well-performing auto-focus algorithm should add minimal computational overhead over the test image acquisition times. We therefore compared DeepFocus processing time to microscope image acquisition time for CPU-only and GPU-based inference, run directly on the microscope control computer. GPU-based inference out-performed CPU-only processing by about an order of magnitude, especially for larger input image patches. Input image patches with an edge length of 384 and 512 pixels allowed fast inference with accurate results (Supplementary Table 1, Fig. 2f). Importantly, DeepFocus processing times did not add substantial overhead, even for the little optimized CPU-only mode (Fig. 2f), which will allow widespread deployment of the algorithm to standard microscope computers without hardware modifications.

We conducted additional tests to investigate whether utilizing more advanced, pretrained image encoders has a beneficial impact on the performance of our approach (Table 1 and Supplementary

Tables 2–5, Supplementary Text 1). Notably, our findings indicate that the pre-trained EfficientNet substantially outperforms the baseline architecture (Stacked Convolutional with an edge length of 512) across all data sets, albeit at the expense of reduced throughput. Using pre-trained weights, the model achieved a considerable reduction in the mean absolute error (MAE) (Supplementary Table 6), closing the gap between EfficientNet and the baseline.

## Learned image region weighting

During DeepFocus development, we noticed that many specimens contain regions with little usable information for an auto-focus algorithm, such as blood vessels in tissue, which show only blank epoxy resin and no contrast that could be used by an auto-focus algorithm (Fig. 3a, b). We therefore reasoned that such areas should have less weight in any $\Delta F$ estimation, and devised a neural network loss term and architecture (Supplementary Fig. 4) that directly leads to the emergence of a second set of model outputs that weigh the $\Delta F$ esti-mates, without additional training data. These new DeepFocus outputs are loosely regularized (only in terms of weight decay) scores that are used as weighting factors already during DeepFocus model training.

**Table 1 | Comparison against common model architectures using the mean absolute error (MAE)**

| Architecture | MAE$_{wd}$ [μm] | MAE$_{stigx}$ [a.u.] | MAE$_{stigy}$ [a.u.] | GPU speed [MPx/s] | CPU speed [MPx/s] |
|---|---|---|---|---|---|
| Stacked Conv. | 2.59 ± 2.39 | 0.45 ± 0.39 | 0.43 ± 0.30 | 518.26 ± 146.90 | 54.00 ± 0.74 |
| ResNet-50 | 3.06 ± 4.24 | 0.52 ± 0.56 | 0.43 ± 0.42 | 102.76 ± 1.24 | 2.60 ± 0.02 |
| EfficientNet-B0 | 1.45 ± 1.43 | 0.29 ± 0.21 | 0.37 ± 0.33 | 406.56 ± 22.25 | 14.14 ± 0.28 |

Single iteration and single input patch-pair performance (mean ± SD) of different model architectures (Stacked Conv. with an edge length of 512) on the neural tissue test data and pixel throughput (Methods).

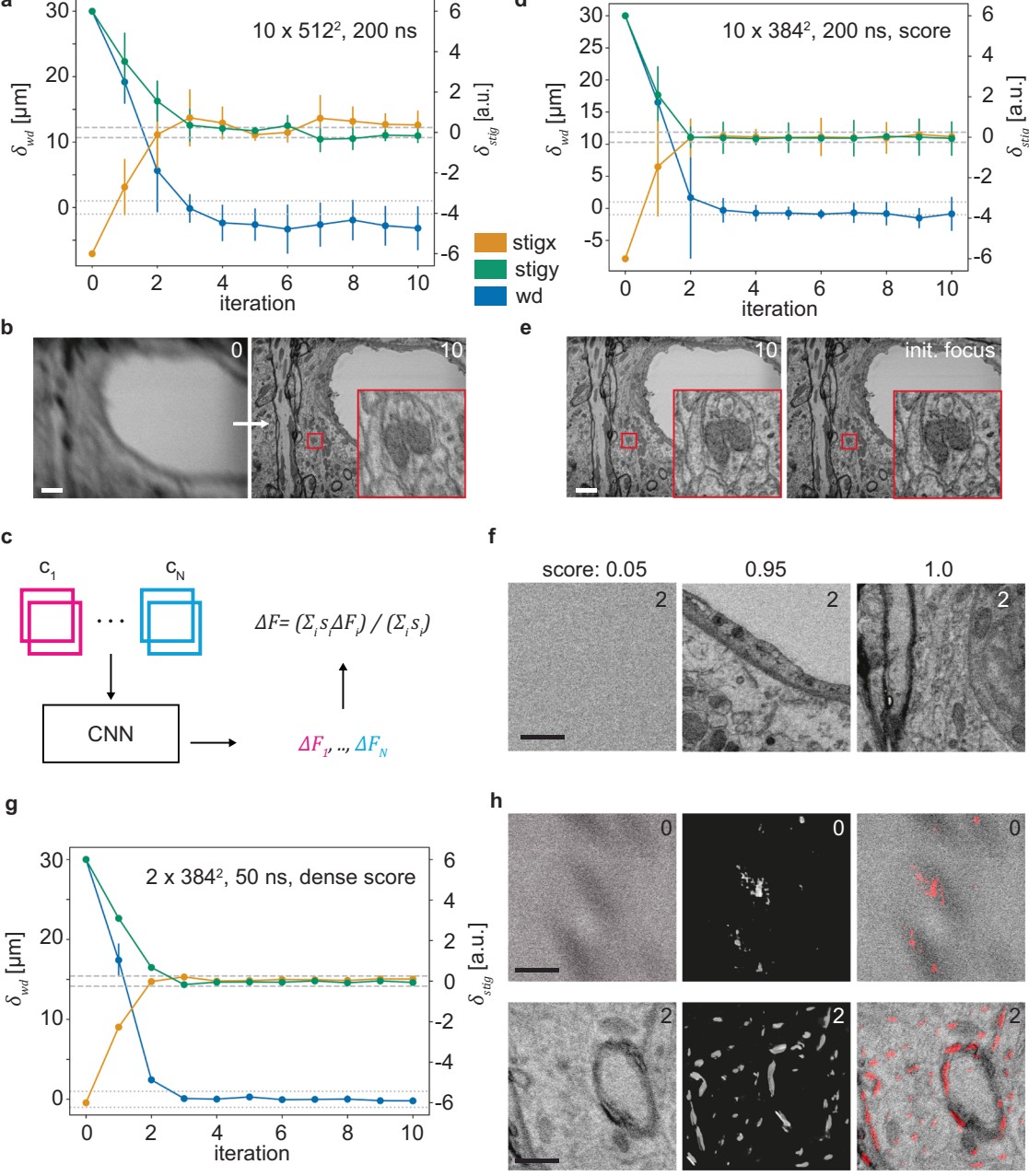

**Fig. 3 | DeepFocus with additional score prediction. a** Convergence of the model from Fig. 2a (using $n = 10$ instead of 5 patch pairs) on the image in **b** that contained a blood vessel; 200 ns dwell time, image size 2048 × 1568. **c** Model architecture that predicts an additional score $s_i$ per patch pair used to calculate a weighted focus correction. **d** Improved convergence of the patch-score model with $n = 10$ input patch pairs (384 × 384). **e** Image after using the score model (left) and the image with correct focus (right). **f** One of the two input patches and corresponding score values ($n = 1$; ratios of maximum values; original values: 0.0065, 0.1235, 0.129).

**g** Convergence with pixel-level score predictions using n = 2 input patch pairs (384 × 384) at 50 ns dwell time and image size 2048 × 1536. **h** Score map of one example patch used in g. The right column shows the composite images of the example input patch (left column) and the corresponding pixel scores (center column) in red. Scale bars: 2 μm in b,e and 0.5 μm for f and h. Data in all plots are presented as mean values ± SD from the n input patches. Source data are provided as a Source Data file.

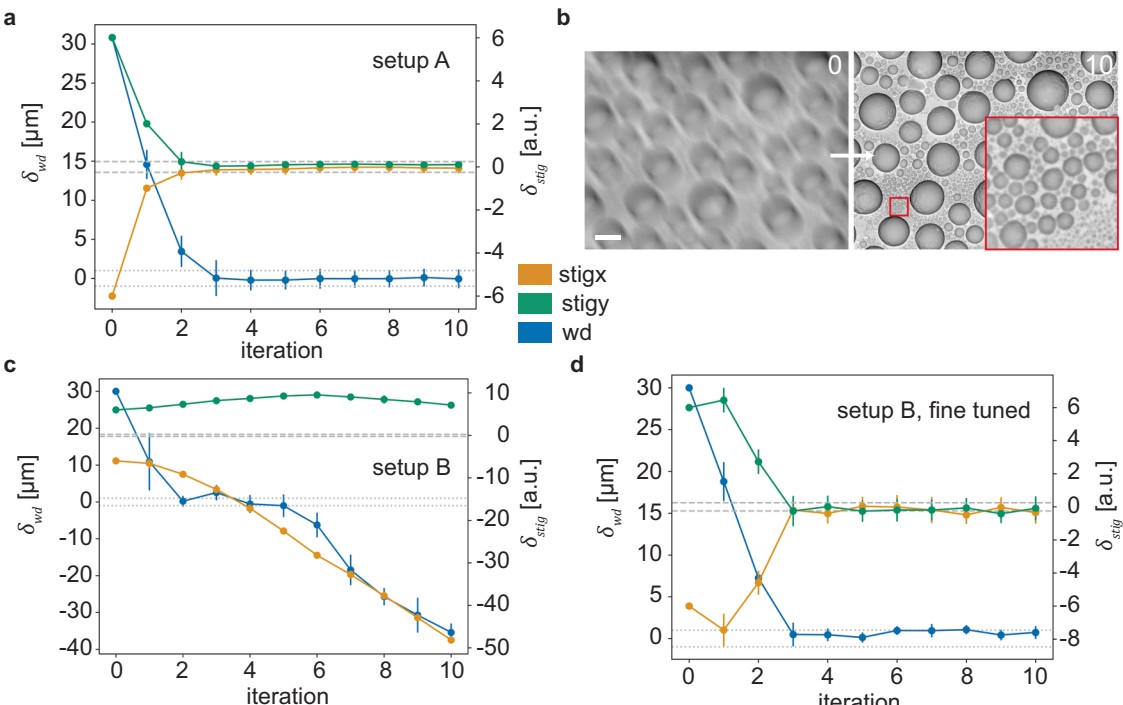

**Fig. 4 | DeepFocus convergence on an unseen sample and recalibration for a different setup. a** Model convergence (same model as in Fig. 3d; $n = 10$ input patch pairs) on a sample of tin on carbon (not contained in the training data) using setup A at 100 ns dwell time. **b** Image from a at iterations 0 and 10. Scale bar is 1 μm. **c** The same model as applied to tin on carbon on setup B (see Methods for imaging parameters). **d** Convergence of the fine-tuned DeepFocus model (last three fully connected layers re-trained) after 50k training iterations using 100 automatically acquired samples at 10 different locations with setup B (Methods). Data in all plots are presented as mean values ± SD. Source data are provided as a Source Data file.

We tested two different granularities for weighting, first on the level of the DeepFocus input image patches (Fig. 3c–f), which are cropped portions of the larger input image pair acquired by the microscope, and second, on the level of individual pixels, leading to a scoring of every location in an input image (Fig. 3g, h). Both approaches proved more robust toward specimen regions with little contrast information, demonstrating that DeepFocus does not require potentially error-prone conventional image processing to pre-filter low-contrast regions.

## Transferability of DeepFocus

Like MAPFoSt (Maximum-A-Posteriori Focusing and Stigmation)[7], several aberration correction algorithms were developed for SEM in the past, and microscope manufacturer software usually includes such algorithms. In our experience, however, these algorithms performed often poorly[7], possibly due to overfitting their parameters, or even the entire algorithmic model to particular test cases. To assess the extent to which DeepFocus is susceptible to overfitting to its remarkably small training set, we first evaluated it on an unseen, non-biological sample and second, on an entirely different microscope, with different imaging settings. Remarkably, DeepFocus generalized exceptionally well to this novel sample (Fig. 4a, b), even with being trained only on image data of a single specimen. Transferring the algorithm to a different microscope with vastly different imaging settings (modified landing energy, beam current, overall working distance range, rotated image acquisition) led to failure and divergence of the model, as expected (Fig. 4c).

## Machine-independent auto focus

We therefore developed an alternative approach, DeepScore, aiming for machine and setting independence, by estimating the magnitude of $\Delta F$ without its correction direction from a single image (Methods,

see[19,20]). The intent was to create a slower yet machine-independent auto-focus algorithm, based on the directionless score and classical optimization (tested with the simplex method developed by Nelder and Mead[21]). This algorithm can then be used to generate, with minimal manual input, a new training set in case of a required DeepFocus re-calibration. We found that DeepScore, when used with classical optimization, can effectively infer a parameter set $F$ that leads to sharp image formation, albeit, as expected, with slower convergence than the regular DeepFocus model (Supplementary Fig. 5, Supplementary Text 2). Using this approach, we generated a new, smaller training data set ($n = 10$ locations, 31% of the original training set) for a SEM where DeepFocus had diverged. Fine tuning the DeepFocus model (recalibration) took less than 2 h on a single GPU, and recovered its ability to estimate $\Delta F$ with the original convergence speed (Fig. 4d).

## Comparison with the MAPFoSt algorithm

We finally performed a direct comparison of DeepFocus and the state-of-the-art automatic aberration correction algorithm for SEM, MAPFoSt. MAPFoSt uses a Bayesian optimal approach to infer the target $\Delta F$ values, and was specifically optimized to yield a parameter set for sharp images with as little electron dose as possible for the sample. We used the publicly available Python implementation of the algorithm (https://pypi.org/project/mapfost/), with parameters adjusted by its developer (RS) for the SEM used. As expected, MAPFoSt was also able to estimate a correct parameter set on the tested samples (Fig. 5, Supplementary Table 7), but required on average 4 more iterations to convergence (residual$_{wd}$ mean and SD of DeepFocus after iteration 2: 0.34 μm ± 0.3 μm vs MAPFoSt after iteration 6: 0.5 μm ± 0.21 μm) despite using 50 ns pixel dwell time for the two perturbed images with DeepFocus, and 200 ns for MAPFoSt. Strikingly, DeepFocus outperforms MAPFoSt in particular for low SNR imagery, the image settings domain it was developed for, and large initial aberrations

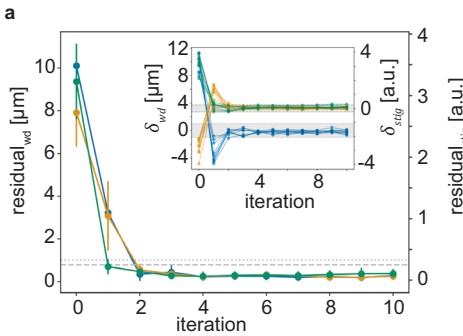
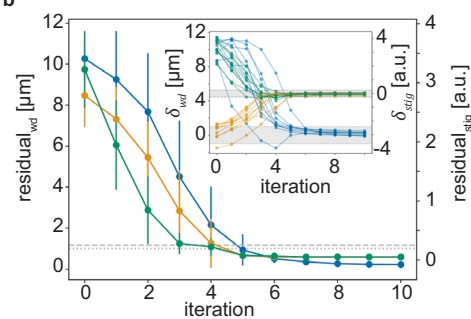

**Fig. 5 | Residual error (mean absolute difference from the baseline, see Methods) of DeepFocus and MAPFoSt using 2048 × 1536 input images.** The individual trajectories are shown in the inset images. **a** DeepFocus model from Fig. 3d with 10 patch pairs (384 × 384) and 50 ns pixel dwell time. **b** MAPFoSt with 4 patch pairs (768 × 768) and 200 ns pixel dwell time. Colors are consistent with Fig. 2a. Dashed and dotted horizontal lines indicate 0.25 and 1 μm margins for the stigmator and working distance, respectively. Data in both plots are presented as mean values ± SD calculated from $n = 9$ trajectories. Source data are provided as a Source Data file.

(Supplementary Fig. 6). We also observed that MAPFoSt required longer computation times, more than 30 times, in comparison to DeepFocus running on a low-power GPU inside the microscope computer (processing time per $512^2$ patch pair with GPU: 0.032 s ± 0.004 s and CPU: 0.240 s ± 0.011 s compared to MAPFoSt with 0.897 s ± 0.024 s for $512^2$ and 1.673 s ± 0.018 s for $768^2$; Methods).

## Discussion

While deep learning has demonstrated impressive advances in recent years in domains such as natural language processing[22] or computer vision[23,24], many "simple" control theory problems remain to be explored[25]. Here we demonstrate how a powerful and over-parameterized model, in the classical sense, can outperform the carefully hand-optimized state-of-the-art approach[7] in all measured performance metrics: robustness toward low SNR images, convergence speed, measured by algorithm iterations, and surprisingly, the calculation duration of inference.

This may not be unexpected, given the success of convolutional neural networks across various domains of computer vision, and the fact that auto-focusing can be framed as a regression problem given two input images with known working distance perturbations. We found that various neural network architectures, from simple convolutional models followed by fully connected layers to more modern U-Nets[26] were able to solve the problem. This suggests that innovation in machine control may shift from carefully crafting models toward carefully connecting and interfacing more general models. It also shows that future versions of DeepFocus will likely perform better, simply by plugging in more powerful standard model architectures.

We believe that the alternative approaches to aberration correction in SEM rely on many implicit and explicit assumptions about the nature of the input images, the electron optics, the point spread function, and, in general, the entire system that is being controlled. While these assumptions are clearly necessary to build a control system based on explicit physical models or classical image processing, they inevitably result in an approximation of the system's behavior. DeepFocus also approximates the system's behavior, through the training data, but with fewer hard assumptions, and leads to an auto-focusing algorithm that is tailored to the peculiarities of every SEM/sample after a simple recalibration, while still generalizing surprisingly well to unseen samples without retraining.

This purely data driven approach also has clear limitations: adjusting the focus under conditions that are underrepresented or not part of the training data (e.g., large magnification changes or very exotic samples) might lead to poor convergence. Whether it will be possible to acquire a training set and train a single model that covers essentially all possible parameters without fine-tuning, also by feeding

microscope parameters directly to the model, remains to be demonstrated.

## Methods

Research performed in this study complies with all relevant ethical regulations of the MPI for Biological Intelligence.

### Electron microscopes and samples

All experiments were performed using two different scanning electron microscopes (SEM). The default setup was a Zeiss Merlin SEM equipped with an in-lens secondary electron detector and operated at an acceleration voltage of 1.5 kV, a beam current of 1.5 nA, and a working distance of 4.5 mm (setup A). Recalibration experiments were carried out on a Zeiss UltraPlus SEM with an acceleration voltage of 1.2 kV, a 60 μm aperture (-0.6 nA beam current), an in-lense secondary electron detector, a working distance of 6 mm, and scans that were rotated by 90° (setup B).

Experiments involving biological samples were conducted on 250 nm sections collected on a silicon wafer. The sections were cut from a 500 μm diameter biopsy punch of a 200 μm thick zebra finch brain slice, stained with Hua protocol[27] and embedded in Spurr's resin. Experiments with non-biological specimens were carried out on tin on carbon from Agar Scientific (S1937) for both setups A and B.

The stigmator values reported by the microscope software (SmartSEM Version 6.06) were used without additional adjustment or calibration.

### Ground truth generation

To generate training and validation samples, a pair of perturbed images (±5 μm working distance) was acquired relative to a known aberration, which was introduced by changing working distance and both stigmators. An expert SEM user manually adjusted the focus baseline at each location, and perturbed image pairs were acquired for 10 introduced aberration vectors (working distance, stig x, stig y). The values of the aberration vectors were drawn uniformly within a given range. Each training sample consisted of two perturbed images as the model input and the corresponding negative aberration vector as the target. The aberrations were sampled at 23 locations from a working distance range of ±20 μm, and astigmatisms of ±0.5. The perturbed images were acquired at a size of 1024 × 768. For 17 locations, images were acquired at a size of 2048 × 1536 and within ±20 μm (wd) and ±5 (stigmators). Finally, the resulting 400 samples were shuffled and divided into training (80%, 320 samples) and validation (20%, 80 samples) sets.

The test data sets for the model architecture comparison comprised 1) neural tissue test data (50 samples, recorded at 5 locations with each 10 aberrations, acquisition with 10 MHz) and 2) tin on carbon

**Table 2 | Stacked convolutional architecture of DeepFocus for 2 x 512 x 512 inputs**

| Layer # | Layer Type | Input Channels | Output Channels | Kernel Size | Pooling Size | Activation | Dropout (p) | Batch Norm. |
|---|---|---|---|---|---|---|---|---|
| 1 | Conv3D | 1 | 20 | (1, 5, 5) | (1, 2, 2) | ReLU | 0.1 | Yes |
| 2 | Conv3D | 20 | 30 | (1, 5, 5) | (1, 2, 2) | ReLU | 0.1 | Yes |
| 3 | Conv3D | 30 | 40 | (1, 4, 4) | (1, 2, 2) | ReLU | 0.1 | Yes |
| 4 | Conv3D | 40 | 50 | (1, 4, 4) | (1, 2, 2) | ReLU | 0.1 | Yes |
| 5 | Conv3D | 50 | 60 | (1, 2, 2) | (1, 2, 2) | ReLU | 0.1 | Yes |
| 6 | Conv3D | 60 | 70 | (1, 1, 1) | (1, 2, 2) | ReLU | 0.1 | Yes |
| 7 | Conv3D | 70 | 70 | (1, 1, 1) | (1, 1, 1) | ReLU | 0.1 | Yes |
| 8 | Linear | 6860 | 250 | - | - | ReLU | - | No |
| 9 | Linear | 250 | 50 | - | - | ReLU | - | No |
| 10 | Linear | 50 | 3 | - | - | - | - | No |

test data (80 samples, 2 locations with 40 aberrations each; one location with 5 MHz and one with 10 MHz). The test images were acquired with aberrations sampled uniformly within ±20 μm (wd) and ±5 (stigmators).

## Model architectures and training

All models were developed and trained using PyTorch[28] 1.9.0 and the open source framework elektronn3 (https://github.com/ELEKTRONN/elektronn3) with mini-batches, $L_1$ loss (mean absolute error), step learning rate scheduler (a factor of 0.99 every 2000 steps) and the AdamW optimizer[29].

The image-to-scalar architecture used seven convolutional layers (valid convolution; 3D kernels to share weights across the two inputs using a z-kernel size of 1) followed by three fully connected layers (Table 2). The convolutional layers were constructed as follows: convolution, batch normalization, activation (ReLU), max-pooling, and dropout (rate $p = 0.1$).

For different input shapes, the parameters of the fully connected layer (rows 8–10 in Table 2) were adjusted as follows:
- 2 × 128 × 128: Linear(140, 100), Linear(100, 50), Linear(50, 3)
- 2 × 256 × 256: Linear(1260, 250), Linear(250, 50), Linear(50, 3)
- 2 × 384 × 384: Linear(3500, 250), Linear(250, 50), Linear(50, 3)

The model output is a correction vector $\Delta\widetilde{F}$ for working distance (in μm) and stig x and y (arbitrary units). The $L_1$ loss was calculated without additional weighting as the value range of the different target types (working distance vs. stigmator) appeared sufficiently similar.

In order to obtain an average estimate of multiple corrections with learned weights, the architecture was modified to produce an output of 4 channels (3 for corrections and a weight score associated with each correction: $\Delta F_i, s_i$) instead of 3. The model was trained by computing the weighted average of 5 predictions using the softmax function for normalization of the scores as weights. During each iteration of the training process, 5 patch pairs were generated from the input, and the resulting model output, which was the weighted average, was compared with the target to calculate the loss.

In the image-to-image case, we employed a 3D U-Net architecture[26] with three planar blocks to facilitate weight sharing between the two input images, same convolution, resize convolutions[30] for the upsampling and group normalization[31]. Our model used 32 start filters and two final 2D conv. layers to project the concatenated channels of the two inputs images to 4 channels per pixel: Conv2D(input channels = 64, output channels = 20, kernel_size = (1, 1)), activation, Conv2D(20, 4, (1, 1)). A softmax function was applied to the 2D score map output which was then used to calculate the weighted average of the per-pixel predictions. Multiple dense predictions were combined by calculating their mean.

In both score models (image-to-scalar and image-to-image) an additional loss term based on the $L_1$ loss of the individual (either patch- or pixel-wise) predictions was added ($\alpha = 0.25$):

$$\widetilde{L} = (1 - \alpha)L_1^{final} + \alpha L_1^{individual} \tag{3}$$

Model inputs (gray scale images with intensities between 0 and 255) were rescaled to -1 and 1. Patch pairs (one for each of the perturbed images) were cropped randomly (but with the same offset; except for the independent version) and augmented (independently applied with probability p; all values were drawn from a Normal distribution) with additive Gaussian noise (p = 0.75, mean=0, sigma=0.2), random gamma adjustment (p = 0.75, mean=1.0, gamma SD = 0.25; pixel intensities internally rescaled between 0 and 1; $I^* = I^p$) and a random brightness and contrast adjustment (contrast mean=1, contrast SD = 0.25, brightness mean = 0, brightness SD = 0.25;

$$I^* = contrast \times = (I - I_{mean}) + I_{mean} + brightness \tag{4}$$

Trainings were stopped after validation loss convergence at $1 \times 10^6$ iterations (no-score models), $0.5 \times 10^6$ (patch-score model) and $0.2 \times 10^6$ (pixel-score model). The number of model parameters (Supplementary Text 1, Supplementary Table 1) and the architecture visualizations in (Supplementary Fig. 4) were retrieved and created with TorchLense[32].

## Convergence experiments

To evaluate the convergence behavior of our models, we monitored the state of the focal parameters during 10 consecutive iterations at a fixed position, using a known initial aberration. Specifically, we plotted the deviation from the focus baseline for three parameters - working distance, stigmator x, and stigmator y - after each iteration. We confirmed that the parameter deviations correlate with improvements in the common image-comparison metrics structural similarity index and mean squared error (Supplementary Table 8).

Experiments with DeepFocus and single trajectories used initial aberrations of (30 μm, −6, 6). To determine the parameter baseline, we first coarsely adjusted the focus manually, and then ran the DeepFocus model with patch scores for three iterations, using a dwell time of 200 ns, an image size of 2048 × 1536, and patches sized at 20 × 384 × 384. We subsequently verified the obtained parameter baseline by visually confirming that it led to sharp images (correct focus values). The signal-to-noise ratio (SNR) of the image presented in Fig. 2 was determined using the methodology proposed by Sage and Unser[33] and a low-noise image, obtained with 800 ns pixel dwell time, as reference (iteration 10 in Fig. 2b, d).

## Compute hardware and timings

Model training was conducted on a Windows computer equipped with two Nvidia Quadro RTX 5000 graphics processing units (GPUs), an Intel Xeon Gold 6240 central processing unit (CPU) @ 2.60 GHz

**Table 3 | DeepScore architecture for 2 x 512 x 512 inputs**

| Layer # | Layer Type | Input Channels | Output Channels | Kernel Size | Pooling Size | Activation | Dropout (p) | BatchNorm. |
|---|---|---|---|---|---|---|---|---|
| 1 | Conv3D | 1 | 20 | (1, 3, 3) | (1, 2, 2) | ReLU | 0.1 | Yes |
| 2 | Conv3D | 20 | 30 | (1, 3, 3) | (1, 2, 2) | ReLU | 0.1 | Yes |
| 3 | Conv3D | 30 | 40 | (1, 3, 3) | (1, 2, 2) | ReLU | 0.1 | Yes |
| 4 | Conv3D | 40 | 50 | (1, 3, 3) | (1, 2, 2) | ReLU | 0.1 | Yes |
| 5 | Conv3D | 50 | 60 | (1, 3, 3) | (1, 2, 2) | ReLU | 0.1 | Yes |
| 6 | Conv3D | 60 | 70 | (1, 3, 3) | (1, 2, 2) | ReLU | 0.1 | Yes |
| 7 | Linear | 2520 | 250 | - | - | ReLU | - | No |
| 8 | Linear | 250 | 50 | - | - | ReLU | - | No |
| 9 | Linear | 50 | 2 | - | - | - | - | No |

(36 threads) and 768 GB RAM. Inference was executed directly on the microscope computers (setup A/B), and the time measurements were carried out on the Zeiss Merlin microscope computer (Intel Xeon CPU E5-2609 v2 @ 2.50 GHz, 4 threads; 16 GB memory; T1000 GPU). The measurements were performed with either the CPU-only or the CUDA (Compute Unified Device Architecture by Nvidia) backend of PyTorch.

The processing time measurement commenced with the perturbed image pair array and concluded with a single correction vector, encompassing cropping, image normalization, CPU-GPU memory transfers, and mean estimation. Initialization of the PyTorch model was excluded from the measurement, as it is only required once during startup. Serialized versions of the model were stored and loaded with TorchScript. The MAPFoSt implementation utilized multithreading on image patches; for instance, for a 2048 × 1536 input image and a patch size of 768 × 768, four parallel processes were spawned. All timing measurements were conducted with 2048 × 1536 images, a pixel dwell time of 200 ns and computed as the mean of 10 repetitions.

**Comparison against common model architectures**

The baseline model (stacked convolutional layers) was compared against ResNet-50[34] and EfficientNet-B0[35] - two common and established architectures. The training of the two reference architectures was performed as described in "Model architectures and training" with initial pre-trained weights (available through PyTorch 'ResNet50_-Weights.IMAGENET1K_V2', 'EfficientNet_B0_Weights.IMAGENET1K_V1'). In order to output a correction vector $\Delta\widetilde{F}$ for working distance (in μm) and stig x and y (arbitrary units), the layers after the average pooling of the original architectures were replaced by two convolutional layers (output channels ResNet: 512 and 4; EfficientNet: 320 and 4; kernel size 1), which received the concatenated results of the two independently processed input patches as input. Both models used 512 × 512 input patches.

The inference speed was measured as pixel throughput with 5 input patch pairs (2 × 512 × 512) cropped from 2048 × 1536 input images and reported as the average of 10 repetitions. The pixel count is calculated as 512 × 512 × 5 × 2. The timings were conducted on the Windows PC described in "Compute hardware and timings".

**Recalibration procedure**

To automatically generate training data for novel setups (DeepFocus recalibration), a separate neural network was developed with the aim of regressing a generalized and microscope-independent image sharpness score (DeepScore). The model designed to produce such a score for a single image was based on an architecture similar to the image-to-scalar DeepFocus variant (Table 3).

The model output comprised two scores: one for the working distance $s_{wd}$ and one for the stigmation $s_{stig}$, which may be used for adjustment later on. The loss was calculated using the $L_1$ distance between the absolute ground truth targets (working distance, stigmator x, stigmator y) and the model outputs. The two, absolute

stigmator components of the ground truth were summed prior to the loss calculation with the model output score $s_{stig}$. To generate a single score per image, the minima of N patch predictions (with locations selected randomly using a fixed initial seed) were computed independently for each score type (working distance and stigmation) and subsequently summed without additional weights. The resulting single score was used for all experiments.

In order to transform the image sharpness score (objective function) into a microscope-independent autofocus algorithm, we combined it with the downhill simplex method[21]. This approach minimizes the DeepScore through iterative adjustment of the focus parameters. We adopted F. Chollet's Python implementation of the Nelder-Mead algorithm (https://github.com/fchollet/nelder-mead), with the following extension: If there was no improvement within the last 5 iterations (at most every 5 iterations), the current focus parameters were perturbed with noise drawn from a uniform distribution within the ranges (±2 μm, ±0.5, ±0.5).

The automatic adjustment of the focus parameter at each location was achieved using the Nelder-Mead-DeepScore autofocus with 10 × 2 × 512 × 512 patches cropped from an input image with a 200 ns pixel dwell time and a resolution of 2048 × 1536 pixels. The DeepScore network was trained on the ground truth acquired on setup A (see Ground truth generation). To derive a threshold to be used as a stopping criterion for the downhill simplex method, the focus was adjusted manually once before initiating the procedure. The corresponding sharpness score was then evaluated and multiplied by 1.05.

The training image pairs for the DeepFocus recalibration on setup B were acquired on a regular grid with a resolution of 2048 × 1536 pixels and a dwell time randomly chosen as either 200 ns or 100 ns. The first 10 locations' samples were used for training, each sampled with 10 aberrations (uniformly drawn between ±20 μm, ±5, ±5; 100 location-aberration pairs in total; stopping threshold 0.0014). Recalibration was then performed by fine-tuning the parameters of the last three fully connected layers of a pre-trained DeepFocus model. Fine-tuning employed the training parameters described for the DeepFocus, except for an increased learning rate decay, which was achieved by multiplying the rate by 0.95 every 1000 steps and limiting training to a maximum of 50,000 steps (approx. 2 h).

**Multi-trajectory recordings and MAPFoSt comparison**

In the experiments conducted with the UltraPlus (setup B) illustrated in Fig. 4, two iterations of the MAPFoSt algorithm (with a 400 ns dwell time and 4 × 786 × 768 patches) were employed to establish the baseline for the unrotated beam scan. Manual focusing, executed by an expert (PS), was utilized for the 90° rotated scan.

Multi-trajectory plots were obtained at 9 distinct locations, evenly distributed on a grid with 80 μm side length. In addition, the mean absolute error (MAE) was computed for each iteration to estimate the average convergence speed and final variance of the model. The initial focus baseline was established through manual focus adjustment,

followed by the application of MAPFoSt twice using a 200 ns dwell time, a resolution of 2048 × 1536, and 768 × 768 patches (Fig. 5a) or the patch-score model (Fig. 5b). This baseline was employed to set the initial aberrations. To account for a minor shift in the target focus (working distance) observed during the final iterations, possibly due to the frequent imaging during the trajectory acquisition, two iterations of MAPFoSt (Fig. 5a) or the patch-score model (in the case of Fig. 5b) were performed post-trajectory recording to obtain a more accurate baseline for plotting trajectories and margins in Fig. 5 and Supplementary Fig. 6a. Patch locations for DeepFocus were chosen randomly, yet with a fixed sequence of seeds, i.e. the same N patch offsets (1 offset per patch pair) were used across all trajectories and iterations. Initial aberrations were uniformly sampled within the following ranges: 8 to 12 μm (working distance), -4 to -2 (stig x), and 2 to 4 (stig y), with a fixed random seed to ensure an identical distribution of aberrations for both MAPFoSt and DeepFocus. The test locations on the specimen for the 9 trajectories were identical for Fig. 5a and Supplementary Fig. 6a. All experiments involving MAPFoSt were conducted with version 4.2.1 (https://pypi.org/project/mapfost/4.2.1/).

### Statistics & Reproducibility

No statistical methods were used to predetermine the sample size. The experiments were not randomized, and the investigators were not blinded to allocation during experiments and outcome assessment.

### Reporting summary

Further information on research design is available in the Nature Portfolio Reporting Summary linked to this article.

## Data availability

All relevant data supporting the key findings of this study are available within the article and its Supplementary Information files. The data generated in this study have been deposited in the Zenodo database under accession code https://doi.org/10.5281/zenodo.8416524[36]. Source data are provided with this paper.

## Code availability

Source code (https://doi.org/10.5281/zenodo.8422791) is publicly available on Zenodo[37].

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

## Acknowledgements

We would like to thank Winfried Denk for generously providing lab resources from his department, and Csilla Pataki and Jonas Hemesath for measuring the Zeiss UltraPlus beam currents. All funding was provided by the Max Planck Society.

## Author contributions

P.S. performed all experiments and conceived the DeepFocus & DeepScore algorithm jointly with J.K. R.S. developed and calibrated the MAPFoSt implementation. The manuscript was written by P.S. & J.K. with contributions by R.S.

## Funding

## Competing interests

A patent application (EP21212051) covering the method described in this manuscript is currently pending. The authors PS and JK are named as inventors. RS declares no competing interests.
