## [Peer Review File · Nature Communications]

DeepFocus: Fast focus and astigmatism correction for electron microscopyReviewer #1 (Remarks to the Author):

This article presents DeepFocus, a simple and fast data-driven algorithm for focus and astigmatism correction of scanning electron microscopy (SEM). DeepFocus has faster converges and less processing time compared with the state-of-the-art method MAPDoSt even on low signal-to-noise ratio images. The results presented in this manuscript are enlightening, but the manuscript still has several weaknesses that are worth noting:

1. The model employed in this manuscript appears to be excessively simplistic and lacks adequate comparison. The model of simply stacking a few 3D convolutions and adding a few fully connected layers lacks comparison with other commonly used backbones such as VGGNet, ResNet, and U-Net. The authors can replace the 2D convolutions in these backbones with 3D convolutions and compare them with the model structure proposed in this manuscript. It can help to validate the superiority of the model structure employed in this manuscript.
2. The experimental results need to include numerical comparisons to enhance clarity. Instead of relying solely on graphs and charts, the authors should present metric results to help readers better understand the specific performance of the model. The authors can utilize metrics such as Peak Signal-to-Noise Ratio (PSNR), Structural Similarity Index (SSIM), Mean Squared Error (MSE), etc, to quantitatively evaluate the in-focus images generated by the DeepFocus and DeepScore methods after controlling the microscope imaging parameters in comparison to the original out-of-focus images. The definitions of these metrics can be found at <https://doi.org/10.1016/j.csbj.2022.04.003>.
3. The authors should make the datasets and codes publicly available during the revision stage to ensure transparency, reproducibility, and promote further research in the field.
4. The model configurations used in the comparative experiments are excessively limited.
 - a) The authors only compare the Bayesian optimal-based MAPFoSt. Although the manuscript mentions that the MAPFoSt is a classic and commonly used method for aberration correction in SEM, the proposed DeepFocus method is based on deep learning, and it is unfair to compare it only with the traditional machine learning algorithm. Instead, it should be evaluated against analogous deep-learning algorithms.
 - b) What are the advantages of DeepFocus compared to deep reinforcement learning like proximal policy optimization and policy gradient?
 - c) In Fig. 5, the curves of both algorithms overlap, making it difficult to compare the specific performance of the DeepFocus and MAPFoSt. The authors should provide some numerical results to present the performance differences between different algorithms more intuitively.
5. Some details of the model are not clear and lack explanations.
 - a) According to the manuscript and the provided pseudocode, the proposed DeepFocus initially applies multiple perturbations to the out-of-focus image. Each perturbation generates two image pairs, which are then stacked into the shape (input channels, number of patch pairs, image height, image width) for input to the model. The final prediction result is obtained by averaging the outputs of different image pairs. If my understanding is correct, I would like to inquire about the rationale behind using 3D convolutions instead of 2D convolutions. Given that the input channels are only 1, wouldn't it be more efficient to treat the number of patch pairs as the channel dimension and input them into a 2D convolution? This approach seems to be more parameter-efficient compared to using 3D convolutions.
 - b) It would be helpful if the authors could explain why use the average of outputs between different image pairs as final predictions. Is it intended to reduce prediction errors? This part should be explained, and if possible, conducting some ablation experiments to validate the effectiveness of this design would be beneficial.
 - c) The mention of the "image-to-image case" on line 293 appears to be unclear. Is it used to provide additional constraints that aid in the training of DeepFocus? In addition, the definition of L1 should also be given in case some readers may not know the meaning of it. The authors should revise the section on model architectures and training to help readers replicate it.
6. The narrative in the article needs further enhancement.
 - a) In the sections discussing model architectures and training, as well as recalibration procedure, it is recommended that the authors present the convolutional layers and fully connected layers using table formats (lines 267-276, lines 279-281, and lines 375-383). The authors may refer to the table format used in this manuscript: <https://doi.org/10.1051/0004-6361/201833648>. The authors can also use images to describe the network structure to enhance the clarity of the manuscript. Additionally, if feasible, including a figure depicting the network architecture would be even more

advantageous. Such a visual representation can significantly enhance the readability of the manuscript.

b) Some figure legends in this manuscript are excessively long and tedious (e.g., Fig. 2 and Fig. 3.). It is recommended that the authors consider subdividing these larger figures directly or reconstructing them by retaining only the essential description and a brief introduction. The explanatory text can then be moved into the main body of the manuscript. This approach will help streamline the figure legends and improve the overall readability of the manuscript.

c) The authors should engage in further discussion regarding the limitations of the proposed method, as well as offering valuable insights into potential avenues for future research.

In conclusion, the ideas presented in the manuscript and the experimental results are enlightening. Therefore, it would be beneficial for the authors to incorporate these comments and revise the manuscript accordingly.

Reviewer #2 (Remarks to the Author):

The authors of DeepFocus aim to use machine learning to automate the process of aberration correction in scanning electron microscopy (SEM). This is a particularly interesting and important application because while electron microscopy is widely used in a variety of fields, data acquisition for biological samples in particular is tedious and time consuming. The authors note that others have tried similar approaches to address independent parameter adjustment for automated microscopy, however these methods are often not able to generalize to new data, instruments or conditions. This paper focuses on simultaneously adjusting several imaging parameters (working distance, and x/y stigmator settings) to improve image quality in a way which can be applied to new systems with minimal changes. Importantly, the image focus/acquisition process is accelerated by a factor of ~ 10 compared to the MAPFoST method.

I think that this work is interesting, and I think that microscopists have shown eagerness to adopt ML into their work. However several changes must be made to this paper before publication.

My first main concern is that the paper does not provide sufficient information to reproduce the results, or to fully understand the process.

- After reading the manuscript, SI, figures, I don't understand how the training produced or how the inference process works. Supp. Fig. 1 shows that the authors start from an optimal image, perturb the image, and then add a second perturbation. Why are two perturbation steps required? Are the ground truth negative ΔF measured relative to the ideal image, or the first perturbation?
- Figures like Fig 2A so the convergence of parameters. How do these iterations work? Is the perturbed image pair fed through the same network multiple times yielding improved results each time? If so, can you comment on why the NN is not able to estimate perturbations in a single iteration if it is trained to preproduce the exact ΔF and not some incremental step?
- This information could be obtained by going through the supplied code in detail, however I think that including a schematic figure to the main text which shows both the training and inference processes would be very helpful to readers.

Secondly, I feel that the paper could be reframed to more clearly highlight improvements over existing methods

- If images are collected with a working distance of 4.5mm, how large of an impact does a perturbation of $\pm 20 \mu\text{m}$ have? Presumably at the beginning of an experiment the working distance would be off by $\gg 20 \mu\text{m}$. How large of a working distance range is this method able to accommodate? What resolution in working distance would human experts consider when focusing an image?
- The goal of collecting clear images is to make post-processing (segmentation, 3D reconstruction, etc.) easier. How sensitive are the available post-processing methods to perturbations of this scale? How different does a 3D reconstruction built from images collected with DeepFocus appear to one constructed from images optimized with MAPFoST, for instance?
- The authors note that computational overhead should be low. This is helpful for maximizing the amount of data that can be collected in a fixed amount of time, however accumulated electron dose on the sample has a huge impact on image quality and sample degradation. Can the author's

comment on how the time for inference compares to the time required to collect 3-5 iterations of images? In a real automated experiment, how often would the instrument need to be refocused?

I was able to run the inference script on my personal computer (MacOS Ventura) without issue. The README file provides sufficient explanation to install the software. There are not many comments in the code, but it is cleanly written and at least as easy to follow as source code for other open-source projects.

Other comments:

- Fig. 3e – what does init. focus refer to? Is it the ideally focused image, or the initial perturbation used to generate training data (like in Supp. Fig. 1). How do the inset images compare? The init focus inset seems to have higher contrast – is this desirable?
- Often in SEM experiments scientists do a fast scan over a small area to focus the optics before collecting a large-scale slow scan. I think this is what the authors are getting emulating by using small image crops with short dwell time, but this could be clarified for the broader audience.
- I'm surprised that the NN training takes 44 hours. There are certainly many parameters, but from my experience in CNN for image segmentation training takes only a few hours for dataset of 1000's of images. Can you comment on this? Would a smaller NN (either in width or depth) perform comparably? What convergence criteria do you aim for that takes so long to achieve?
- You have clearly shown that the model is able to generalize, but can you comment on overfitting with a small dataset?
- Can you comment on how unique suggested aberration parameters are? Are the multiple combinations of stig-x, stig-y, and wd that could produce the same image?

Reviewer #3 (Remarks to the Author):

DeepFocus: Fast focus and astigmatism correction for electron microscopy

The manuscript describes a data-driven method for fast focusing and aberrations corrections in scanning electron microscopy (SEM). The development of methods such as the one presented by the authors is of great importance in the field of electron microscopy in many different aspects; imaging of beam-sensitive materials and large volumes imaging both in 2D and 3D all can benefit from fast autocorrections to the focus and astigmatism. SEMs can be used to image any solid material that is synthetically produced or can be found on Earth. Organic samples and mostly biological samples are relatively more complicated to image as they are beam sensitive and have low contrast. The authors present the DeepFocus method and demonstrate how they train it and apply it to stained biological samples. In comparison to other available methods, DeepFocus seems to be faster and easy to apply to other instruments and potentially to other samples. I think that it is a well-written and important manuscript, and it should be published but I do have some concerns about the generalization of the method and I'd like to ask the authors to address a few topics.

Here are my comments and questions regarding the application of the DeepFocus method in SEM imaging:

- 1- SEMs can be operated in almost endless combinations of landing/accelerating voltages apertures/currents/spot size, working distances, detectors, and with different samples that produce variable intensities of signals and contrasts. Can you please mention if you tried to change any SEM parameter such as landing voltage or aperture/current? Are there any limitations?
- 2- The authors mentioned their great success working with low signal-to-noise ratios. The dwell times reported in the manuscript are indeed very short and result in low SNR ratios, but in both SEMs, the current or aperture that was used to produce the images is relatively large. To the best of my knowledge, SEM imaging and especially beam-sensitive samples as biological samples cannot withstand such high currents and are damaged very quickly, usually the smallest aperture (10-20 μ m) or currents of tens pA are used to minimize the damage. Lower currents will result in lower SNR, can the authors suggest if DeepFocus will be able to successfully perform under such conditions?
- 3- What about magnifications or the horizontal field of view, are there any limitations? Any recommended ranges?
- 4- Did the authors try DeepFocus on other biological samples, for instance, frozen samples under cryo conditions, without any staining? Can you please comment on the possibility to use it in cryo-

SEM, I believe that DeepFocus is of great importance to these challenging beam-sensitive, low-contrast samples.

5- In the methods section, lines 234-240, the imaging parameters in the Zeiss Merlin are described in landing voltage and beam current, and in the Zeiss Ultra Plus they are described in landing voltage and aperture size. Please change the parameters to be the same, voltage and current.

Response to referees

Responses from the authors are in *italics*.

Reviewer #1 (Remarks to the Author):

This article presents DeepFocus, a simple and fast data-driven algorithm for focus and astigmatism correction of scanning electron microscopy (SEM). DeepFocus has faster converges and less processing time compared with the state-of-the-art method MAPDoSt even on low signal-to-noise ratio images. The results presented in this manuscript are enlightening, but the manuscript still has several weaknesses that are worth noting:

1. The model employed in this manuscript appears to be excessively simplistic and lacks adequate comparison. The model of simply stacking a few 3D convolutions and adding a few fully connected layers lacks comparison with other commonly used backbones such as VGGNet, ResNet, and U-Net. The authors can replace the 2D convolutions in these backbones with 3D convolutions and compare them with the model structure proposed in this manuscript. It can help to validate the superiority of the model structure employed in this manuscript.

We appreciate the reviewer's constructive feedback. In response, we have conducted additional experiments using different model architectures. We would also like to clarify that we had already used a U-Net for the image-to-image case, as mentioned in the methods: "In the image-to-image case, we employed a 3D U-Net architecture."

Interestingly, we observed that our initial model, conceived more as a simple proof-of-concept for DeepFocus rather than a product of an extensive architecture search, was surpassed by a pre-trained EfficientNet on our test data. This suggests that the potential of the DeepFocus method could be even greater with further enhanced architectures in the future.

2. The experimental results need to include numerical comparisons to enhance clarity. Instead of relying solely on graphs and charts, the authors should present metric results to help readers better understand the specific performance of the model. The authors can utilize metrics such as Peak Signal-to-Noise Ratio (PSNR), Structural Similarity Index (SSIM), Mean Squared Error (MSE), etc, to quantitatively evaluate the in-focus images generated by the DeepFocus and DeepScore methods after controlling the microscope imaging parameters in comparison to the original out-of-focus images. The definitions of these metrics can be found at <https://doi.org/10.1016/j.csbj.2022.04.003>.

We would like to highlight that we have already utilized an appropriate metric, MAE, as described in the methods section. For enhanced clarity, we have introduced Table 1, which now also features the results from the additional model comparisons we conducted. Furthermore, we have provided a table comparing the metrics SSIM and MSE between the baseline (initial focus) and the image obtained after iterative model application (Supp. Table 7).

3. The authors should make the datasets and codes publicly available during the revision stage to ensure transparency, reproducibility, and promote further research in the field.

We have made datasets and code available on GitHub, already before publication, as requested.

4. The model configurations used in the comparative experiments are excessively limited.

a) The authors only compare the Bayesian optimal-based MAPFoSt. Although the manuscript mentions that the MAPFoSt is a classic and commonly used method for aberration correction in SEM, the proposed DeepFocus method is based on deep learning, and it is unfair to compare it only with the traditional machine learning algorithm. Instead, it should be evaluated against analogous deep-learning algorithms.

While we agree that additional comparisons are generally advantageous, it is pertinent to note that when MAPFoSt was introduced, extensive comparisons with the most common autofocusing method for our microscopes (the proprietary and closed-source algorithm provided by the manufacturer) were performed, wherein MAPFoSt exhibited superior performance. Additionally, as the source code for other algorithms is not readily available, incorporating them would necessitate potentially month- or even year-long experiments, which, from our perspective, are out-of-scope for this study. Nonetheless, we have conducted further comparisons with other pretrained model architectures.

b) What are the advantages of DeepFocus compared to deep reinforcement learning like proximal policy optimization and policy gradient?

DeepFocus employs training with both direct supervised and self-supervised loss terms, typically making it more efficient than reinforcement learning approaches. Our problem formulation, which is the core innovation, enables this efficient training.

c) In Fig. 5, the curves of both algorithms overlap, making it difficult to compare the specific performance of the DeepFocus and MAPFoSt. The authors should provide some numerical results to present the performance differences between different algorithms more intuitively.

We have now added Table 1 (and Supp. Tables 1-6) that provide these numerical results more intuitively. In none of our experiments, MAPFoSt outperformed DeepFocus.

5. Some details of the model are not clear and lack explanations.

a) According to the manuscript and the provided pseudocode, the proposed DeepFocus initially applies multiple perturbations to the out-of-focus image. Each perturbation generates two image pairs, which are then stacked into the shape (input channels, number of patch pairs, image height, image width) for input to the model. The final prediction result is obtained by averaging the outputs of different image pairs. If my understanding is correct, I would like to inquire about the rationale behind using 3D

convolutions instead of 2D convolutions. Given that the input channels are only 1, wouldn't it be more efficient to treat the number of patch pairs as the channel dimension and input them into a 2D convolution? This approach seems to be more parameter-efficient compared to using 3D convolutions.

Thanks for the interesting line of thought. Indeed, stacking the two input patches along the channel axis would increase the total number of model parameters. With 3D convolutions, convolution kernels are shared between the two input patches (with a kernel size of 1 in the third dimension). However, the parameter count does see a slight increase with the concatenation approach due to the augmented number of input channels. Our rationale behind using 3D convolutions, with shared parameters/kernels, was to maintain distinct transformations for each of the two patches. This approach aims to compel the formation of kernels that independently extract a proxy of the beam parameters for each patch. The desired model output, the beam parameter "difference," is then determined in the last layers.

b) It would be helpful if the authors could explain why use the average of outputs between different image pairs as final predictions. Is it intended to reduce prediction errors? This part should be explained, and if possible, conducting some ablation experiments to validate the effectiveness of this design would be beneficial.

We added Supp. Table 3 which contains the model performances when using the average of 10 predictions (10 pairs) instead of 1 (Table 1). This simple, yet effective consensus strategy allows to 1) provide more robust prediction results and 2) estimate the spread of the predictions, which might be useful during "production" to detect uncertain prediction results.

c) The mention of the "image-to-image case" on line 293 appears to be unclear. Is it used to provide additional constraints that aid in the training of DeepFocus? In addition, the definition of L1 should also be given in case some readers may not know the meaning of it. The authors should revise the section on model architectures and training to help readers replicate it.

The image-to-image approach is an extension of the patch-based method. In this approach, the model learns to determine which parts of the image are informative and which are not, thanks to the introduction of an additional loss term (see Methods under "Model architectures and training"). To enhance clarity, we have revised these sections. Additionally, we are providing the full source code and datasets, which should simplify replication considerably.

6. The narrative in the article needs further enhancement.

a) In the sections discussing model architectures and training, as well as recalibration procedure, it is recommended that the authors present the convolutional layers and fully connected layers using table formats (lines 267-276, lines 279-281, and lines 375-383). The authors may refer to the table format used in this manuscript: <https://doi.org/10.1051/0004-6361/201833648>. The authors can also use images to describe the network structure to enhance the clarity of the manuscript. Additionally, if feasible, including a figure depicting the network architecture would be even more advantageous. Such a visual representation can significantly enhance the readability of the manuscript.

We have followed the suggestion and provide tables that describe the layers, as well as added Supp. Fig. 4 that provides detailed network architecture visualizations using the torchlense (<https://www.nature.com/articles/s41598-023-40807-0>) package.

b) Some figure legends in this manuscript are excessively long and tedious (e.g., Fig. 2 and Fig. 3.). It is recommended that the authors consider subdividing these larger figures directly or reconstructing them by retaining only the essential description and a brief introduction. The explanatory text can then be moved into the main body of the manuscript. This approach will help streamline the figure legends and improve the overall readability of the manuscript.

We have shortened and improved the clarity of the excessively long and tedious figure legends 2 and 3, thanks for the suggestion.

c) The authors should engage in further discussion regarding the limitations of the proposed method, as well as offering valuable insights into potential avenues for future research.

In light of this feedback and comments from reviewer 3, we have expanded the discussion on the limitations of our current approach. Furthermore, we have mentioned that future versions of DeepFocus could potentially perform even better, simply by integrating improved standard model backbones expected to emerge in the coming years.

In conclusion, the ideas presented in the manuscript and the experimental results are enlightening. Therefore, it would be beneficial for the authors to incorporate these comments and revise the manuscript accordingly.

We would like to thank the reviewer again for the thoughtful comments that allowed us to improve the manuscript and method substantially.

Reviewer #2 (Remarks to the Author):

The authors of DeepFocus aim to use machine learning to automate the process of aberration correction in scanning electron microscopy (SEM). This is a particularly interesting and important application because while electron microscopy is widely used in a variety of fields, data acquisition for biological samples in particular is tedious and time consuming. The authors note that others have tried similar approaches to address independent parameter adjustment for automated microscopy, however these methods are often not able to generalize to new data, instruments or conditions. This paper focuses on simultaneously adjusting several imaging parameters (working distance, and x/y stigmator settings) to improve image quality in a way which can be applied to new systems with minimal changes. Importantly, the image focus/acquisition process is accelerated by a factor of ~10 compared to the MAPFoST method.

I think that this work is interesting, and I think that microscopists have shown eagerness to adopt ML into their work. However several changes must be made to this paper before publication.

My first main concern is that the paper does not provide sufficient information to reproduce the results, or to fully understand the process.

- After reading the manuscript, SI, figures, I don't understand how the training produced or how the inference process works. Supp. Fig. 1 shows that the authors start from an optimal image, perturb the image, and then add a second perturbation. Why are two perturbation steps required? Are the ground truth negative DF measured relative to the ideal image, or the first perturbation?

We have made the full source code and dataset available through a GitHub repository. This should make it easier for others to reproduce our results. In addition, the following explanations, combined with the revisions in the manuscript, aim to clarify our approach further:

- *Two perturbation steps are essential to allow the neural network to gauge not only the magnitude but also the direction of the necessary parameter adjustments. For example, in the context of defocus, a perturbation of the working distance/focus by +1 μm centered around current values would yield just a single, blurry image. Although the network can determine if this image is in focus and estimate the defocus magnitude (as discussed in the 'DeepScore' section of the manuscript), it doesn't provide adequate information for precise parameter adjustment.*
- *Ground truth is determined in relation to the parameters that define the "ideal image".*

- Figures like Fig 2A so the convergence of parameters. How do these iterations work? Is the perturbed image pair fed through the same network multiple times yielding improved results each time? If so, can you comment on why the NN is not able to estimate perturbations in a single iteration if it is trained to reproduce the exact DF and not some incremental step?

Indeed, in some cases, the network necessitates multiple iterations or runs. Each run uses updated images from the microscope, obtained based on the outcomes of the preceding iteration, to achieve a fully in-focus image -- particularly when dealing with significant initial aberrations. We theorize that estimating these large initial aberrations in a singular step is quite challenging. However, we believe the network adequately generalizes to predict the correct direction for an update.

- This information could be obtained by going through the supplied code in detail, however I think that including a schematic figure to the main text which shows both the training and inference processes would be very helpful to readers.

We agree that this was not yet very clear and have updated the main Fig. 1 to indicate that multiple iterations can be required in the inference process. During training, only "single steps" are trained, which

is shown in Supp. Fig. 1 - we would prefer to keep this figure in the supplements though, as it takes up a lot of space.

Secondly, I feel that the paper could be reframed to more clearly highlight improvements over existing methods

- If images are collected with a working distance of 4.5mm, how large of an impact does a perturbation of +/- 20 mm have? Presumably at the beginning of an experiment the working distance would be off by >> 20 mm. How large of a working distance range is this method able to accommodate? What resolution in working distance would human experts consider when focusing an image?

We have evaluated working distance deviations from the optimal focus plane of up to 30 μm (we assume +/- 20 mm refers to 20 μm) and these images appear already completely blurry. Human experts, according to our measurements in Fig. 1 b, cannot resolve differences less than 1.0 μm in working distance well.

- The goal of collecting clear images is to make post-processing (segmentation, 3D reconstruction, etc.) easier. How sensitive are the available post-processing methods to perturbations of this scale? How different does a 3D reconstruction built from images collected with DeepFocus appear to one constructed from images optimized with MAPFoST, for instance?

The 3d reconstructions built from images collected with DeepFocus should ideally appear identically to those reconstructed from images taken with MAPFoST, as both algorithms are capable of finding parameters that lead to good image quality. The key advantages of DeepFocus are much faster processing and convergence.

- The authors note that computational overhead should be low. This is helpful for maximizing the amount of data that can be collected in a fixed amount of time, however accumulated electron dose on the sample has a huge impact on image quality and sample degradation. Can the author's comment on how the time for inference compares to the time required to collect 3-5 iterations of images? In a real automated experiment, how often would the instrument need to be refocused?

Panel 2f shows how the time for inference compares to the time required to collect the images for a single iteration, which does not change for multiple iterations. How often the instrument requires refocusing depends highly on sample stability, temperature stability and other properties, and the time between necessary refocusing can range in our experience from many seconds (very unstable conditions) to multiple hours (very stable conditions).

I was able to run the inference script on my personal computer (MacOS Ventura) without issue. The README file provides sufficient explanation to install the software. There are not many comments in the code, but it is cleanly written and at least as easy to follow as source code for other open-source projects.

Thanks a lot for the positive feedback.

Other comments:

- Fig. 3e – what does init. focus refer to? Is it the ideally focused image, or the initial perturbation used to generate training data (like in Supp. Fig. 1). How do the inset images compare? The init focus inset seems to have higher contrast – is this desirable?

Init. focus refers to the ideally focused image (on the right in Fig. 3e), which is compared to an image after running 10 iterations of the improved autofocus model that was trained on ignoring regions without much information (here the bloodvessel) which could be used to estimate the parameters. The slightly higher contrast might be an artefact of repeated imaging of the same region for the experiment.

- Often in SEM experiments scientists do a fast scan over a small area to focus the optics before collecting a large-scale slow scan. I think this is what the authors are getting emulating by using small image crops with short dwell time, but this could be clarified for the broader audience.

This is correct - it also limits the total dose on the sample and makes it faster to autofocus, as long as the autofocus algorithm can make use of the limited information (low SNR due to fast scanning and small area).

- I'm surprised that the NN training takes 44 hours. There are certainly many parameters, but from my experience in CNN for image segmentation training takes only a few hours for dataset of 1000's of images. Can you comment on this? Would a smaller NN (either in width or depth) perform comparably? What convergence criteria do you aim for that takes so long to achieve?

It's worth noting that many neural networks dedicated to image segmentation, particularly those achieving top-tier performance, are trained for months on large GPU clusters. Our usual goal is to achieve a visually flat training and validation loss curve. We've included comparisons with larger models, which indeed demonstrate superior performance, thus we're not inclined to reduce the model size. Considering the enhancement achieved with pre-trained model weights (sourced from ImageNet), we'd like to emphasize that a foundational model tailored for EM data could further amplify both model performance and training efficiency.

- You have clearly shown that the model is able to generalize, but can you comment on overfitting with a small dataset?

The largest model tested now (ResNet), shows signs of overfitting, which is not entirely surprising as our training set is small. In case DeepFocus gets integrated by microscope manufacturers into their device control software, we expect that larger training sets will likely be of use.

- Can you comment on how unique suggested aberration parameters are? Are the multiple combinations of stig-x, stig-y, and wd that could produce the same image?

This is correct, which is why we need two perturbations to estimate the correction vector.

Reviewer #3 (Remarks to the Author):

DeepFocus: Fast focus and astigmatism correction for electron microscopy

The manuscript describes a data-driven method for fast focusing and aberrations corrections in scanning electron microscopy (SEM). The development of methods such as the one presented by the authors is of great importance in the field of electron microscopy in many different aspects; imaging of beam-sensitive materials and large volumes imaging both in 2D and 3D all can benefit from fast autocorrections to the focus and astigmatism. SEMs can be used to image any solid material that is synthetically produced or can be found on Earth. Organic samples and mostly biological samples are relatively more complicated to image as they are beam sensitive and have low contrast. The authors present the DeepFocus method and demonstrate how they train it and apply it to stained biological samples. In comparison to other available methods, DeepFocus seems to be faster and easy to apply to other instruments and potentially to other samples. I think that it is a well-written and important manuscript, and it should be published but I do have some concerns about the generalization of the method and I'd like to ask the authors to address a few topics.

We would like to thank the reviewer for the positive feedback.

Here are my comments and questions regarding the application of the DeepFocus method in SEM imaging:

1- SEMs can be operated in almost endless combinations of landing/accelerating voltages apertures/currents/spot size, working distances, detectors, and with different samples that produce variable intensities of signals and contrasts. Can you please mention if you tried to change any SEM parameter such as landing voltage or aperture/current? Are there any limitations?

This is correct, the parameter combination space is indeed vast. Fundamentally, we believe that as long as human operators can produce a focused image, DeepFocus should function effectively. Nevertheless, this might necessitate model retraining, as illustrated in Fig. 4. The feasibility of training a singular DeepFocus model that can generalize across all potential conditions and samples is still uncertain. Such an endeavor would likely mandate the development of an extensive training/test set, which was outside the scope of this manuscript. Moreover, it might prove beneficial or even essential to inform the model with the complete "parameter state space" of the microscope -- another intriguing consideration we had but did not explore within this paper's scope. We primarily view this study as a proof-of-principle demonstration, laying the foundation for future refinements. To convey this perspective more effectively, we've incorporated a 'limitations' section in the discussion.

2- The authors mentioned their great success working with low signal-to-noise ratios. The dwell times reported in the manuscript are indeed very short and result in low SNR ratios, but in both SEMs, the

current or aperture that was used to produce the images is relatively large. To the best of my knowledge, SEM imaging and especially beam-sensitive samples as biological samples cannot withstand such high currents and are damaged very quickly, usually the smallest aperture (10-20mm) or currents of tens pA are used to minimize the damage. Lower currents will result in lower SNR, can the authors suggest if DeepFocus will be able to successfully perform under such conditions?

This is correct, we usually use the microscopes with large currents and short dwell times, to rapidly scan large areas. We expect it to perform well as long as humans can recognize structures somewhat clearly (min. SNR range of 1-2), but whether it will operate under extremely low SNR conditions, which usually requires integration of multiple images to extract information, has to be tested.

3- What about magnifications or the horizontal field of view, are there any limitations? Any recommended ranges?

We have not conducted extensive tests on magnification changes, nor have we explored using patches from the corners of large images (which might exhibit distortions) or spanning significant pixel size ranges. Our efforts have primarily centered around simple up- and downsampling of images before inputting them into the model, which proved satisfactory within the 10 nm to 50 nm range. Substantial changes would presumably necessitate a more comprehensive training set and might also call for the inclusion of microscope parameters as additional model input, as mentioned before. We view this as beyond the current manuscript's scope and would like to leave this to future research.

4- Did the authors try DeepFocus on other biological samples, for instance, frozen samples under cryo conditions, without any staining? Can you please comment on the possibility to use it in cryo-SEM, I believe that DeepFocus is of great importance to these challenging beam-sensitive, low-contrast samples.

While we have no experience with cryo-EM ourselves (this is not our research domain), we indeed see no fundamental limitations of our proposed approach wrt cryo-EM, especially with an adjusted training set, and agree that DeepFocus could help this technology in the future.

5- In the methods section, lines 234-240, the imaging parameters in the Zeiss Merlin are described in landing voltage and beam current, and in the Zeiss Ultra Plus they are described in landing voltage and aperture size. Please change the parameters to be the same, voltage and current.

The Zeiss UltraPlus uses an aperture system to adjust the beam current, while the Zeiss Merlin allows continuous adjustment and to directly set the current. We have measured the beam currents on the Zeiss UltraPlus microscope manually and report the values additionally in the manuscript.

Reviewer #1 (Remarks to the Author):

Thank you for addressing my previous comments. I appreciate your responses, as you have greatly improved the readability and comprehensiveness of the submitted manuscript. However, I still found a few issues in this article that require further improvement:

Since this article focuses on the application of DeepFocus, it would be beneficial for the author to conduct additional tests related to the scalability of DeepFocus. In practical model deployment, storage space is often limited. In addition to testing DeepFocus for inference speed with different input sizes under a specific backbone network, it is also essential to assess the impact of different input sizes on model parameters and prediction accuracy. The author could continue testing DeepFocus when using the StackedConv proposed in this article as the backbone network, testing how different input sizes affects the model parameter size and prediction accuracy of StackedConv. In the end, the author can provide a setting that achieves the optimal trade-off between inference speed, parameter size, and prediction accuracy.

The text and details in the article need further refinement as there are numerous minor mistakes. For example, in line 370, "x" should be used instead of "*". In lines 324 to 325, "." should not be used to represent multiplication as it can be confusing. In line 354, "x" should not be used to denote multiplication. The authors use four different symbols (i.e., "x," "x," ".", "*") to represent the same operation for multiplication is highly undesirable. The author should ensure that the same meaning of operations is represented using the same symbol to avoid confusion for the readers. Furthermore, the font in figures and legends should be consistent with the main text. For instance, variables like σ_{wd} and ΔF in the main text are in italics, but Figure 1 and figure legends do not follow this convention. There are many similar errors throughout the entire article, and I cannot point out each one individually. The author should perform a comprehensive review to enhance the article's readability and consistency.

In summary, I appreciate the author's revisions in this round, and I hope the author will continue to refine the article to improve its quality.

Reviewer #2 (Remarks to the Author):

The revised version of DeepFocus: Fast focus and astigmatism correction for electron microscopy is much improved over the previous version. I think that the manuscript should be accepted for publication, but I have a few suggestions to improve clarity.

1. In my previous review I questioned the training procedure, specifically why two perturbations are needed. The authors provided a clear explanation in their response/rebuttal, but this was not included in the manuscript itself. Understanding how the data set was built and how the model is trained is key for understanding the results. Nature Communications has a very broad audience, and while this perturbation step seems to be used in other auto-focusing methods (as described in the paragraph starting at line 32) this may not be common knowledge for all readers. I think it would be helpful to say explicitly in the manuscript that multiple perturbations are needed to determine both the magnitude and direction of parameter adjustments.

2. I don't fully understand Figure 2e. The plot showing a linear relationship between predicted and target working distance corrections is clear and it is clearly describe in the text. However, the test for stigx and stigy is not clear. To the best of my understanding, the authors aim to test performance over a large range of defocus with no stig perturbation, and they are trying to show that the DeepFocus model learns to adjust only the correct parameter and not to compensate for defocus by adjusting stigx/stigy. This needs to be described in the text - the current statement "...the initially unaltered stigmator parameters were minimally affected" is unclear since the stigx/stigy parameters were not addressed at all in the previous paragraph (starting at line 87).

Reviewer #3 (Remarks to the Author):

The authors had answered successfully all my comments.

Round 2:

REVIEWER COMMENTS

Reviewer #1 (Remarks to the Author):

Thanks for putting in the time to help us improve the manuscript!

Thank you for addressing my previous comments. I appreciate your responses, as you have greatly improved the readability and comprehensiveness of the submitted manuscript. However, I still found a few issues in this article that require further improvement:

Since this article focuses on the application of DeepFocus, it would be beneficial for the author to conduct additional tests related to the scalability of DeepFocus. In practical model deployment, storage space is often limited. In addition to testing DeepFocus for inference speed with different input sizes under a specific backbone network, it is also essential to assess the impact of different input sizes on model parameters and prediction accuracy. The author could continue testing DeepFocus when using the StackedConv proposed in this article as the backbone network, testing how different input sizes affects the model parameter size and prediction accuracy of StackedConv. In the end, the author can provide a setting that achieves the optimal trade-off between inference speed, parameter size, and prediction accuracy.

We again appreciate the reviewer's constructive feedback and have conducted additional experiments using different input sizes, with results reported in Supp. Table 1. Based on our measurements in Fig. 2f and Supp. Table 1, we conclude that input sizes of 384 and 512 are a good trade-off, and report this in the manuscript.

The text and details in the article need further refinement as there are numerous minor mistakes. For example, in line 370, "x" should be used instead of "*". In lines 324 to 325, "." should not be used to represent multiplication as it can be confusing. In line 354, "x" should not be used to denote multiplication. The authors use four different symbols (i.e., "x", "x", ".", "**") to represent the same operation for multiplication is highly undesirable. The author should ensure that the same meaning of operations is represented using the same symbol to avoid confusion for the readers. Furthermore, the font in figures and legends should be consistent with the main text. For instance, variables like σ_{wd} and ΔF in the main text are in italics, but Figure 1 and figure legends do not follow this convention. There are many similar errors throughout the entire article, and I cannot point out each one individually. The author should perform a comprehensive review to enhance the article's readability and consistency.

We again appreciate the feedback and have worked on improving the consistency w.r.t. to the use of symbols and other formatting issues. We hope that any remaining minor issues in that regard can be addressed by the copy editors.

In summary, I appreciate the author's revisions in this round, and I hope the author will continue to refine the article to improve its quality.

Reviewer #2 (Remarks to the Author):

Thanks for putting in the time to help us improve the manuscript!

The revised version of DeepFocus: Fast focus and astigmatism correction for electron microscopy is much improved over the previous version. I think that the manuscript should be accepted for publication, but I have a few suggestions to improve clarity.

1. In my previous review I questioned the training procedure, specifically why two perturbations are needed. The authors provided a clear explanation in their response/rebuttal, but this was not included in the manuscript itself. Understanding how the data set was built and how the model is trained is key for understanding the results. Nature Communications has a very broad audience, and while this perturbation step seems to be used in other auto-focusing methods (as described in the paragraph starting at line 32) this may not be common knowledge for all readers. I think it would be helpful to say explicitly in the manuscript that multiple perturbations are needed to determine both the magnitude and direction of parameter adjustments.

Thanks for pointing this out, we have included this now explicitly in the manuscript "The DeepFocus algorithm takes as input two SEM images ... to estimate the direction and magnitude correction of the beam parameters, exploiting phase diversity. Note, that a single image is not sufficient to estimate the aberrations."

2. I don't fully understand Figure 2e. The plot showing a linear relationship between predicted and target working distance corrections is clear and it is clearly describe in the text. However, the test for stigx and stigy is not clear. To the best of my understanding, the authors aim to test performance over a large range of defocus with no stig perturbation, and they are trying to show that the DeepFocus model learns to adjust only the correct parameter and not to compensate for defocus by adjusting stigx/stigy. This needs to be described in the text - the current statement "...the initially unaltered stigmator parameters were minimally affected" is unclear since the stigx/stigy parameters were not addressed at all in the previous paragraph (starting at line 87).

This is correct, and our description was indeed not very helpful, thanks for pointing this out. We have addressed this in the main text: "Notably, the stigmator values were barely changed by the model, when just the working distance was perturbed (Fig. 2e, Supplementary Fig. 3)." We also reference Supplementary Fig. 3 earlier, as it shows the barely changing stigmator values for defocus-only correction at a better scale.

Reviewer #3 (Remarks to the Author):

The authors had answered successfully all my comments.

Thanks for putting in the time to help us improve the manuscript!

Round 1:

Response to referees

Responses from the authors are in *italics*.

Reviewer #1 (Remarks to the Author):

This article presents DeepFocus, a simple and fast data-driven algorithm for focus and astigmatism correction of scanning electron microscopy (SEM). DeepFocus has faster converges and less processing time compared with the state-of-the-art method MAPDoSt even on low signal-to-noise ratio images. The results presented in this manuscript are enlightening, but the manuscript still has several weaknesses that are worth noting:

1. The model employed in this manuscript appears to be excessively simplistic and lacks adequate comparison. The model of simply stacking a few 3D convolutions and adding a few fully connected layers lacks comparison with other commonly used backbones such as VGGNet, ResNet, and U-Net. The authors can replace the 2D convolutions in these backbones with 3D convolutions and compare them with the model structure proposed in this manuscript. It can help to validate the superiority of the model structure employed in this manuscript.

We appreciate the reviewer's constructive feedback. In response, we have conducted additional experiments using different model architectures. We would also like to clarify that we had already used a U-Net for the image-to-image case, as mentioned in the methods: "In the image-to-image case, we employed a 3D U-Net architecture."

Interestingly, we observed that our initial model, conceived more as a simple proof-of-concept for DeepFocus rather than a product of an extensive architecture search, was surpassed by a pre-trained EfficientNet on our test data. This suggests that the potential of the DeepFocus method could be even greater with further enhanced architectures in the future.

2. The experimental results need to include numerical comparisons to enhance clarity. Instead of relying solely on graphs and charts, the authors should present metric results to help readers better understand the specific performance of the model. The authors can utilize metrics such as Peak Signal-to-Noise Ratio (PSNR), Structural Similarity Index (SSIM), Mean Squared Error (MSE), etc, to quantitatively evaluate the in-focus images generated by the DeepFocus and DeepScore methods after controlling the microscope imaging parameters in comparison to the original out-of-focus images. The definitions of these metrics can be found at <https://doi.org/10.1016/j.csbj.2022.04.003>.

We would like to highlight that we have already utilized an appropriate metric, MAE, as described in the methods section. For enhanced clarity, we have introduced Table 1, which now also features the results from the additional model comparisons we conducted. Furthermore, we have provided a table comparing the metrics SSIM and MSE between the baseline (initial focus) and the image obtained after iterative model application (Supp. Table 7).

3. The authors should make the datasets and codes publicly available during the revision stage to ensure transparency, reproducibility, and promote further research in the field.

We have made datasets and code available on GitHub, already before publication, as requested.

4. The model configurations used in the comparative experiments are excessively limited.

a) The authors only compare the Bayesian optimal-based MAPFoSt. Although the manuscript mentions that the MAPFoSt is a classic and commonly used method for aberration correction in SEM, the proposed DeepFocus method is based on deep learning, and it is unfair to compare it only with the traditional machine learning algorithm. Instead, it should be evaluated against analogous deep-learning algorithms.

While we agree that additional comparisons are generally advantageous, it is pertinent to note that when MAPFoSt was introduced, extensive comparisons with the most common autofocusing method for our microscopes (the proprietary and closed-source algorithm provided by the manufacturer) were performed, wherein MAPFoSt exhibited superior performance. Additionally, as the source code for other algorithms is not readily available, incorporating them would necessitate potentially month- or even year-long experiments, which, from our perspective, are out-of-scope for this study. Nonetheless, we have conducted further comparisons with other pretrained model architectures.

b) What are the advantages of DeepFocus compared to deep reinforcement learning like proximal policy optimization and policy gradient?

DeepFocus employs training with both direct supervised and self-supervised loss terms, typically making it more efficient than reinforcement learning approaches. Our problem formulation, which is the core innovation, enables this efficient training.

c) In Fig. 5, the curves of both algorithms overlap, making it difficult to compare the specific performance of the DeepFocus and MAPFoSt. The authors should provide some numerical results to present the performance differences between different algorithms more intuitively.

We have now added Table 1 (and Supp. Tables 1-6) that provide these numerical results more intuitively. In none of our experiments, MAPFoSt outperformed DeepFocus.

5. Some details of the model are not clear and lack explanations.

a) According to the manuscript and the provided pseudocode, the proposed DeepFocus initially applies multiple perturbations to the out-of-focus image. Each perturbation generates two image pairs, which are then stacked into the shape (input channels, number of patch pairs, image height, image width) for input to the model. The final prediction result is obtained by averaging the outputs of different image pairs. If my understanding is correct, I would like to inquire about the rationale behind using 3D convolutions instead of 2D convolutions. Given that the input channels are only 1, wouldn't it be more efficient to treat the number of patch pairs as the channel dimension and input them into a 2D convolution? This approach seems to be more parameter-efficient compared to using 3D convolutions.

Thanks for the interesting line of thought. Indeed, stacking the two input patches along the channel axis would increase the total number of model parameters. With 3D convolutions, convolution kernels are shared between the two input patches (with a kernel size of 1 in the third dimension). However, the parameter count does see a slight increase with the concatenation approach due to the augmented number of input channels. Our rationale behind using 3D convolutions, with shared parameters/kernels, was to maintain distinct transformations for each of the two patches. This approach aims to compel the formation of kernels that independently extract a proxy of the beam parameters for each patch. The desired model output, the beam parameter "difference," is then determined in the last layers.

b) It would be helpful if the authors could explain why use the average of outputs between different image pairs as final predictions. Is it intended to reduce prediction errors? This part should be explained, and if possible, conducting some ablation experiments to validate the effectiveness of this design would be beneficial.

We added Supp. Table 3 which contains the model performances when using the average of 10 predictions (10 pairs) instead of 1 (Table 1). This simple, yet effective consensus strategy allows to 1) provide more robust prediction results and 2) estimate the spread of the predictions, which might be useful during "production" to detect uncertain prediction results.

c) The mention of the "image-to-image case" on line 293 appears to be unclear. Is it used to provide additional constraints that aid in the training of DeepFocus? In addition, the definition of L1 should also be given in case some readers may not know the meaning of it. The authors should revise the section on model architectures and training to help readers replicate it.

The image-to-image approach is an extension of the patch-based method. In this approach, the model learns to determine which parts of the image are informative and which are not, thanks to the introduction of an additional loss term (see Methods under "Model architectures and training"). To enhance clarity, we have revised these sections. Additionally, we are providing the full source code and datasets, which should simplify replication considerably.

6. The narrative in the article needs further enhancement.

a) In the sections discussing model architectures and training, as well as recalibration procedure, it is recommended that the authors present the convolutional layers and fully connected layers using table formats (lines 267-276, lines 279-281, and lines 375-383). The authors may refer to the table format used in this manuscript: <https://doi.org/10.1051/0004-6361/201833648>. The authors can also use images to describe the network structure to enhance the clarity of the manuscript. Additionally, if feasible, including a figure depicting the network architecture would be even more advantageous. Such a visual representation can significantly enhance the readability of the manuscript.

We have followed the suggestion and provide tables that describe the layers, as well as added Supp. Fig. 4 that provides detailed network architecture visualizations using the torchlense (<https://www.nature.com/articles/s41598-023-40807-0>) package.

b) Some figure legends in this manuscript are excessively long and tedious (e.g., Fig. 2 and Fig. 3.). It is recommended that the authors consider subdividing these larger figures directly or reconstructing them by retaining only the essential description and a brief introduction. The explanatory text can then be moved into the main body of the manuscript. This approach will help streamline the figure legends and improve the overall readability of the manuscript.

We have shortened and improved the clarity of the excessively long and tedious figure legends 2 and 3, thanks for the suggestion.

c) The authors should engage in further discussion regarding the limitations of the proposed method, as well as offering valuable insights into potential avenues for future research.

In light of this feedback and comments from reviewer 3, we have expanded the discussion on the limitations of our current approach. Furthermore, we have mentioned that future versions of DeepFocus could potentially perform even better, simply by integrating improved standard model backbones expected to emerge in the coming years.

In conclusion, the ideas presented in the manuscript and the experimental results are enlightening. Therefore, it would be beneficial for the authors to incorporate these comments and revise the manuscript accordingly.

We would like to thank the reviewer again for the thoughtful comments that allowed us to improve the manuscript and method substantially.

Reviewer #2 (Remarks to the Author):

The authors of DeepFocus aim to use machine learning to automate the process of aberration correction in scanning electron microscopy (SEM). This is a particularly interesting and important application because while electron microscopy is widely used in a variety of fields, data acquisition for biological

samples in particular is tedious and time consuming. The authors note that others have tried similar approaches to address independent parameter adjustment for automated microscopy, however these methods are often not able to generalize to new data, instruments or conditions. This paper focuses on simultaneously adjusting several imaging parameters (working distance, and x/y stigmator settings) to improve image quality in a way which can be applied to new systems with minimal changes. Importantly, the image focus/acquisition process is accelerated by a factor of ~10 compared to the MAPFoST method.

I think that this work is interesting, and I think that microscopists have shown eagerness to adopt ML into their work. However several changes must be made to this paper before publication.

My first main concern is that the paper does not provide sufficient information to reproduce the results, or to fully understand the process.

- After reading the manuscript, SI, figures, I don't understand how the training produced or how the inference process works. Supp. Fig. 1 shows that the authors start from an optimal image, perturb the image, and then add a second perturbation. Why are two perturbation steps required? Are the ground truth negative DF measured relative to the ideal image, or the first perturbation?

We have made the full source code and dataset available through a GitHub repository. This should make it easier for others to reproduce our results. In addition, the following explanations, combined with the revisions in the manuscript, aim to clarify our approach further:

- *Two perturbation steps are essential to allow the neural network to gauge not only the magnitude but also the direction of the necessary parameter adjustments. For example, in the context of defocus, a perturbation of the working distance/focus by $+1\ \mu\text{m}$ centered around current values would yield just a single, blurry image. Although the network can determine if this image is in focus and estimate the defocus magnitude (as discussed in the 'DeepScore' section of the manuscript), it doesn't provide adequate information for precise parameter adjustment.*
- *Ground truth is determined in relation to the parameters that define the "ideal image".*

- Figures like Fig 2A so the convergence of parameters. How do these iterations work? Is the perturbed image pair fed through the same network multiple times yielding improved results each time? If so, can you comment on why the NN is not able to estimate perturbations in a single iteration if it is trained to reproduce the exact DF and not some incremental step?

Indeed, in some cases, the network necessitates multiple iterations or runs. Each run uses updated images from the microscope, obtained based on the outcomes of the preceding iteration, to achieve a fully in-focus image -- particularly when dealing with significant initial aberrations. We theorize that estimating these large initial aberrations in a singular step is quite challenging. However, we believe the network adequately generalizes to predict the correct direction for an update.

- This information could be obtained by going through the supplied code in detail, however I think that including a schematic figure to the main text which shows both the training and inference processes would be very helpful to readers.

We agree that this was not yet very clear and have updated the main Fig. 1 to indicate that multiple iterations can be required in the inference process. During training, only “single steps” are trained, which is shown in Supp. Fig. 1 - we would prefer to keep this figure in the supplements though, as it takes up a lot of space.

Secondly, I feel that the paper could be reframed to more clearly highlight improvements over existing methods

- If images are collected with a working distance of 4.5mm, how large of an impact does a perturbation of +/- 20 mm have? Presumably at the beginning of an experiment the working distance would be off by >> 20 mm. How large of a working distance range is this method able to accommodate? What resolution in working distance would human experts consider when focusing an image?

We have evaluated working distance deviations from the optimal focus plane of up to 30 μm (we assume +/- 20 mm refers to 20 μm) and these images appear already completely blurry. Human experts, according to our measurements in Fig. 1 b, cannot resolve differences less than 1.0 μm in working distance well.

- The goal of collecting clear images is to make post-processing (segmentation, 3D reconstruction, etc.) easier. How sensitive are the available post-processing methods to perturbations of this scale? How different does a 3D reconstruction built from images collected with DeepFocus appear to one constructed from images optimized with MAPFoST, for instance?

The 3d reconstructions built from images collected with DeepFocus should ideally appear identically to those reconstructed from images taken with MAPFoST, as both algorithms are capable of finding parameters that lead to good image quality. The key advantages of DeepFocus are much faster processing and convergence.

- The authors note that computational overhead should be low. This is helpful for maximizing the amount of data that can be collected in a fixed amount of time, however accumulated electron dose on the sample has a huge impact on image quality and sample degradation. Can the author's comment on how the time for inference compares to the time required to collect 3-5 iterations of images? In a real automated experiment, how often would the instrument need to be refocused?

Panel 2f shows how the time for inference compares to the time required to collect the images for a single iteration, which does not change for multiple iterations. How often the instrument requires refocusing depends highly on sample stability, temperature stability and other properties, and the time

between necessary refocusing can range in our experience from many seconds (very unstable conditions) to multiple hours (very stable conditions).

I was able to run the inference script on my personal computer (MacOS Ventura) without issue. The README file provides sufficient explanation to install the software. There are not many comments in the code, but it is cleanly written and at least as easy to follow as source code for other open-source projects.

Thanks a lot for the positive feedback.

Other comments:

- Fig. 3e – what does init. focus refer to? Is it the ideally focused image, or the initial perturbation used to generate training data (like in Supp. Fig. 1). How do the inset images compare? The init focus inset seems to have higher contrast – is this desirable?

Init. focus refers to the ideally focused image (on the right in Fig. 3e), which is compared to an image after running 10 iterations of the improved autofocus model that was trained on ignoring regions without much information (here the bloodvessel) which could be used to estimate the parameters. The slightly higher contrast might be an artefact of repeated imaging of the same region for the experiment.

- Often in SEM experiments scientists do a fast scan over a small area to focus the optics before collecting a large-scale slow scan. I think this is what the authors are getting emulating by using small image crops with short dwell time, but this could be clarified for the broader audience.

This is correct - it also limits the total dose on the sample and makes it faster to autofocus, as long as the autofocus algorithm can make use of the limited information (low SNR due to fast scanning and small area).

- I'm surprised that the NN training takes 44 hours. There are certainly many parameters, but from my experience in CNN for image segmentation training takes only a few hours for dataset of 1000's of images. Can you comment on this? Would a smaller NN (either in width or depth) perform comparably? What convergence criteria do you aim for that takes so long to achieve?

It's worth noting that many neural networks dedicated to image segmentation, particularly those achieving top-tier performance, are trained for months on large GPU clusters. Our usual goal is to achieve a visually flat training and validation loss curve. We've included comparisons with larger models, which indeed demonstrate superior performance, thus we're not inclined to reduce the model size. Considering the enhancement achieved with pre-trained model weights (sourced from ImageNet), we'd like to emphasize that a foundational model tailored for EM data could further amplify both model performance and training efficiency.

- You have clearly shown that the model is able to generalize, but can you comment on overfitting with a small dataset?

The largest model tested now (ResNet), shows signs of overfitting, which is not entirely surprising as our training set is small. In case DeepFocus gets integrated by microscope manufacturers into their device control software, we expect that larger training sets will likely be of use.

- Can you comment on how unique suggested aberration parameters are? Are the multiple combinations of stig-x, stig-y, and wd that could produce the same image?

This is correct, which is why we need two perturbations to estimate the correction vector.

Reviewer #3 (Remarks to the Author):

DeepFocus: Fast focus and astigmatism correction for electron microscopy

The manuscript describes a data-driven method for fast focusing and aberrations corrections in scanning electron microscopy (SEM). The development of methods such as the one presented by the authors is of great importance in the field of electron microscopy in many different aspects; imaging of beam-sensitive materials and large volumes imaging both in 2D and 3D all can benefit from fast autocorrections to the focus and astigmatism. SEMs can be used to image any solid material that is synthetically produced or can be found on Earth. Organic samples and mostly biological samples are relatively more complicated to image as they are beam sensitive and have low contrast. The authors present the DeepFocus method and demonstrate how they train it and apply it to stained biological samples. In comparison to other available methods, DeepFocus seems to be faster and easy to apply to other instruments and potentially to other samples. I think that it is a well-written and important manuscript, and it should be published but I do have some concerns about the generalization of the method and I'd like to ask the authors to address a few topics.

We would like to thank the reviewer for the positive feedback.

Here are my comments and questions regarding the application of the DeepFocus method in SEM imaging:

1- SEMs can be operated in almost endless combinations of landing/accelerating voltages apertures/currents/spot size, working distances, detectors, and with different samples that produce variable intensities of signals and contrasts. Can you please mention if you tried to change any SEM parameter such as landing voltage or aperture/current? Are there any limitations?

This is correct, the parameter combination space is indeed vast. Fundamentally, we believe that as long as human operators can produce a focused image, DeepFocus should function effectively. Nevertheless, this might necessitate model retraining, as illustrated in Fig. 4. The feasibility of training a singular DeepFocus model that can generalize across all potential conditions and samples is still uncertain. Such

an endeavor would likely mandate the development of an extensive training/test set, which was outside the scope of this manuscript. Moreover, it might prove beneficial or even essential to inform the model with the complete "parameter state space" of the microscope -- another intriguing consideration we had but did not explore within this paper's scope. We primarily view this study as a proof-of-principle demonstration, laying the foundation for future refinements. To convey this perspective more effectively, we've incorporated a 'limitations' section in the discussion.

2- The authors mentioned their great success working with low signal-to-noise ratios. The dwell times reported in the manuscript are indeed very short and result in low SNR ratios, but in both SEMs, the current or aperture that was used to produce the images is relatively large. To the best of my knowledge, SEM imaging and especially beam-sensitive samples as biological samples cannot withstand such high currents and are damaged very quickly, usually the smallest aperture (10-20mm) or currents of tens pA are used to minimize the damage. Lower currents will result in lower SNR, can the authors suggest if DeepFocus will be able to successfully perform under such conditions?

This is correct, we usually use the microscopes with large currents and short dwell times, to rapidly scan large areas. We expect it to perform well as long as humans can recognize structures somewhat clearly (min. SNR range of 1-2), but whether it will operate under extremely low SNR conditions, which usually requires integration of multiple images to extract information, has to be tested.

3- What about magnifications or the horizontal field of view, are there any limitations? Any recommended ranges?

We have not conducted extensive tests on magnification changes, nor have we explored using patches from the corners of large images (which might exhibit distortions) or spanning significant pixel size ranges. Our efforts have primarily centered around simple up- and downsampling of images before inputting them into the model, which proved satisfactory within the 10 nm to 50 nm range. Substantial changes would presumably necessitate a more comprehensive training set and might also call for the inclusion of microscope parameters as additional model input, as mentioned before. We view this as beyond the current manuscript's scope and would like to leave this to future research.

4- Did the authors try DeepFocus on other biological samples, for instance, frozen samples under cryo conditions, without any staining? Can you please comment on the possibility to use it in cryo-SEM, I believe that DeepFocus is of great importance to these challenging beam-sensitive, low-contrast samples.

While we have no experience with cryo-EM ourselves (this is not our research domain), we indeed see no fundamental limitations of our proposed approach wrt cryo-EM, especially with an adjusted training set, and agree that DeepFocus could help this technology in the future.

5- In the methods section, lines 234-240, the imaging parameters in the Zeiss Merlin are described in landing voltage and beam current, and in the Zeiss Ultra Plus they are described in landing voltage and aperture size. Please change the parameters to be the same, voltage and current.

The Zeiss UltraPlus uses an aperture system to adjust the beam current, while the Zeiss Merlin allows continuous adjustment and to directly set the current. We have measured the beam currents on the Zeiss UltraPlus microscope manually and report the values additionally in the manuscript.

Reviewer #1 (Remarks to the Author):

The authors had successfully addressed all my comments.